# The impact of wildfire on biogeochemical fluxes and water quality on boreal catchments

Gustaf Granath[1], Christopher D. Evans[2,3], Joachim Strengbom[4], Jens Fölster[3], Achim Grelle[4], Johan Strömqvist[5], Stephan J. Köhler[3]

5   [1]Department Ecology and Genetics, Uppsala University, Norbyvägen 18D, Uppsala, Sweden
[2]Centre for Ecology and Hydrology, Bangor, LL57 2UW, UK.
[3]Department of aquatic sciences and assessment, Swedish university of agricultural sciences, Box 7050, Uppsala, Sweden.
[4]Department of Ecology, Swedish University of Agricultural Sciences, Box 7044. SE-750 07 Uppsala, Sweden.
[5]Swedish Meteorological and Hydrological Institute (SMHI), SE-601 76 Norrköping, Sweden.

10   *Correspondence to*: Gustaf Granath (gustaf.granath@gmail.com)

**Abstract.** Wildfires are the major disturbance in boreal ecosystems, and are of great importance for the biogeochemical cycles of carbon (C) and nutrients. However, these fire-induced impacts are hard to quantify and rarely assessed together at an ecosystem level incorporating both aquatic and terrestrial environments. Following a wildfire in Sweden in an area with ongoing monitoring, we conducted a pre-(9 yrs) and post-fire (4 yrs) multi-catchment investigation of element losses 15 (combustion and leaching), and impacts on water quality. Direct C and nitrogen (N) losses through combustion were ca. 4500 g m$^{-2}$ and 100 g m$^{-2}$, respectively. Net $CO_2$ loss associated with soil and biomass respiration was ~150 g C m$^{-2}$ during the first year, but the ecosystem started to show net $CO_2$ uptake in June three years post-fire. Aquatic C and N losses the first 12 months post-fire were 7 g m$^{-2}$ and 0.6 g m$^{-2}$, respectively. Hence, soil respiration comprised a non-negligible part of the post-fire C loss, whereas aquatic C losses were minor, and did not increase post-fire. However, other elements (e.g., Ca, S) 20 exhibited ecologically relevant increases in fluvial export and concentration, with large peaks in the immediate post-fire period. The temporal dynamics of stream concentrations ($Ca^{2+}$, $Mg^{2+}$, $K^+$, $SO_4^{-2}$, $Cl^-$, $NH_4^+$, Total organic N) suggest the presence of faster- and slower-release nutrient pools with half-lives of around 2 weeks and 4 months, which we attribute to physicochemically and biologically mediated mobilisation processes, respectively. Three years after the fire, it appears that dissolved fluxes of nutrients have largely returned to pre-fire conditions, but there is still net release of $CO_2$.

25   **1 Introduction**

Wildfires are the major disturbance agent in boreal ecosystems and are expected to increase in size and frequency (Flannigan et al., 2009). Wildfires have a large impact on biogeochemical cycles, and emissions of $CO_2$ to the atmosphere from more frequent and larger wildfires could generate a positive climate feedback unless the carbon (C) emitted is swiftly re-sequestered (Bond-Lamberty et al., 2007; Smithwick et al., 2005). Wildfires also influence the biogeochemical cycles of 30   nitrogen (N) and major cations (Brais et al., 2000; Grier, 1975; Smithwick et al., 2005), which can influence post-fire

ecosystem productivity, an issue which has been discussed for decades (e.g. Ahlgren and Ahlgren, 1960; Grier, 1975). Losses occur both as emissions during the fire and through post-fire losses via runoff. However, these fire-induced impacts are hard to quantify, and are rarely assessed at an ecosystem level including both aquatic and terrestrial environments (Amiro et al., 2010; Brais et al., 2000; Rhoades et al., 2018; Turner et al., 2007). Comparing post-fire responses to pre-fire conditions is also problematic, because wildfires rarely take place at locations with pre-fire measurements. Here we present a unique pre- and post-fire multi-catchment investigation of water quality and element cycling in boreal Sweden.

Boreal wildfires often consume a large portion of the fuel in form of ground vegetation, and can also consume the upper organic soil (Amiro et al., 2000; Turetsky et al., 2011). Up to 90% of the emitted carbon typically comes from the organic soil layer and in North America, such C emissions are estimated to be on average 3000-4000 g C m$^{-2}$ (Turetsky et al., 2011; Walker et al., 2018). In drained peatlands, the increased exposure of organic soil to oxygen means that C losses can be one order of magnitude larger than uplands and undrained peatlands (Granath et al., 2016). In addition to C, N is also emitted in large quantities during fires (Johnson et al., 2007) as it starts to volatilise at 200°C (Knicker, 2007). This contrasts to other nutrients (e.g. K, P) that require a combustion temperature above 760°C (Knicker, 2007), which rarely occurs. Although N losses can potentially influence long-term ecosystem productivity (Tamm, 1991), few studies have quantified N emissions via this pathway (Brais et al., 2000; Johnson et al., 2007). Studies that have quantified ecosystem C and N emitted during wildfires are still scarce, and are lacking for Northern Europe, impeding our understanding of how wildfires alter major geochemical cycles.

Boreal wildfires do not only cause direct emissions of C and nutrients, but can also alter their fluvial transport and thus downstream water quality (Bladon et al., 2014). To what extent this is true for C does, however, depend on the compound measured, catchment characteristics, and probably fire severity (Santos et al., 2019). Studies have shown negative, little or no effect on the total amount of dissolved organic carbon (DOC) exported postfire (see discussion in Evans et al., 2017: Rodríguez-Cardona et al., 2020), whereas DOC aromaticity and particulate organic carbon (POC) export can increase (Burd et al., 2018; Evans et al., 2017; Olefeldt et al., 2013). More striking is the increase of available macronutrients and other elements that are released from the burnt organic top layer. Typically, the loss of soil cation exchange capacity resulting from combustion of organic soil, together with combustion of biomass, leads to the release of exchangeable cations (e.g. $Ca^{2+}$, $Mg^{2+}$, and $K^+$; González-Pérez et al., 2004). These ions are easily exported to streams and lakes, and can lead to an increase in runoff pH. On the other hand, many studies have shown post-fire peaks in sulphate ($SO_4^{2-}$), chloride ($Cl^-$) and nitrate ($NO_3^-$) due to a combination of release from soil and reduced biological demand (notably for $NO_3^-$) (Bayley et al., 1992; Bladon et al., 2008; Carignan et al., 2000; Lydersen et al., 2014; Mast and Clow, 2008). If acid anions ($NO_3^-$, $SO_4^{2-}$ and $Cl^-$) dominate over base cations, an acidity effect is observed in downstream waters (Lydersen et al., 2014). This acidification effect is enhanced in areas which have higher concentrations of stored S or N from historic deposition, or have a high proportion of peatlands (Bayley et al., 1992; Evans et al., 2017). Lower pH increases dissolved P in the post-fire soil

(Certini, 2005) and a long-term (3-5 years) increase in exported P in burned catchments has been reported across boreal Canada (Burd et al., 2018; Burke et al., 2005; Lamontagne et al., 2000; Silins et al., 2014). However, a high base cation concentration may counterbalance the downstream acidity effect (Carignan et al., 2000).

Nitrogen levels in runoff water normally increase dramatically post-fire (e.g. Bladon et al., 2008; Carignan et al., 2000). Following fire, soil organic nitrogen is either volatilised, or converted into ammonium ($NH_4^+$), while nitrate ($NO_3^-$) is mainly formed from $NH_4^+$ through nitrification, a process which can continue for several years after the fire (Certini, 2005). With the loss of vegetation after a severe fire, and limited potential for microbial immobilization due to a shortage of labile carbon, ammonium and nitrate cannot be retained within the ecosystem and are commonly leached out (Smith et al., 2011). Nitrate

concentrations peak shortly after the fire, but the return time to reference values seems to vary from two to 9 years post-fire (e.g. Bladon et al., 2008; Carignan et al., 2000; Evans et al., 2017; Hauer and Spencer, 1998; Mast and Clow, 2008). In contrast to $NO_3^-$, $NH_4^+$ is expected to be held by the soil to a higher degree because it adsorbed onto negatively charged surfaces of soil particles (Mroz et al., 1980). However, a study observed $NH_4^+$ pulses that lasted over 2 growing seasons (Grogan et al., 2000).

Variation in surface water quality and fluvial transport in a boreal catchment is mainly controlled by landscape heterogeneity (Humborg et al., 2004). For example, the proportion of peatlands in a catchment has a major influence on surface water DOC and $NO_3^-$, which affect runoff pH through the release of organic acids (Buffam et al., 2007; Sponseller et al., 2014). Peatlands naturally retain sulphur under waterlogged conditions (in reduced organic forms and sulphides), so wildfires may

lead to particularly high $SO_4^{2-}$ leaching when peatlands burn. Beside peatlands, lakes upstream can act as buffers in the system by increasing residence time. This will dampen the water quality response to wildfire at the catchment outlet, and possibly reduce the biogeochemical signal via element retention (e.g. in sediments). Despite the clear effect of landscape characteristics on water chemistry, we currently know little about what determines the magnitude or temporal dynamics of post-fire element leaching at the landscape scale.

In 2014, a large wildfire affecting established monitoring sites in Sweden created the opportunity to study ecosystem-level effects of wildfire on biogeochemical cycles in a managed boreal landscape. Whole-catchment studies are important in ecosystem science (Likens et al., 1970) but difficult to conduct at a detailed level, particularly in relation to unpredictable events such as wildfires. In our study, the burnt area (circa 13 000 ha) consists of multiple catchments, allowing us to

investigate local variation in post-fire responses. One of the catchment streams and one lake are included in the Swedish national water monitoring network, enabling comparison with pre-fire data, and with longer-term trends in water chemistry. Hence, compared to most studies, our study does not rely on a single catchment or only post-fire data (see Betts and Jones, 2009; Evans et al., 2017; Mast et al., 2016 for other before-after studies). In addition, it is rarely possible to study

biogeochemical processes during the critical period immediately following a fire, due to limited access to the area, as well as resource constraints.

The overarching aim of this study is to examine the impact of wildfire on element fluxes and water quality in boreal forests.
For element fluxes, we **(i)** estimate carbon and N losses through combustion during the fire and the amount of hydrologically exported C, N, S, Ca, K the first three years post-fire, **(ii)** relate these losses to pre-fire element pools to determine the importance of fire on element cycling fluxes in boreal systems, and **(iii),** estimate the C balance of the system over three years by comparing the C losses with net fluxes and examine plant regrowth. For water quality, we **(i)** describe post fire water quality trends in five streams and one lake, **(ii)** analyse three years of post fire water quality and relate peaks to ten
years of pre-fire baseline data from one stream and one lake within the burnt area, and **(iii)**, determine the form of element concentration decay curves (single or double exponential decay curves, Minderman 1968) to understand post-fire biogeochemical cycling and ecosystem recovery.

## 2 Materials and Methods

### 2.1 Study area

The study area is boreal forest located in Southern Sweden (59°54'50"N, 16°09'50"E). It is located about 75 to 150 m above the sea level that has a low relief but is topographically complex. Between 1987–2016 the mean annual temperature was 6°C (January −3.3°C, July 17°C) and the annual precipitation was 687 mm. The forest is intensively managed using clearcutting, planting, and thinning operations that create a mix of even-aged forest stands from recently cut areas to mature stands (>100 years). Tree cover is dominated by *Pinus sylvestris* (particularly the catchments investigated here), shrub layer by *Vaccinium*
*myrtillus*, *V. vitis-idea*, *Calluna vulgaris*, *Rhododendron tomentosum*, and ground layer by *Pleurozium schreberi*, *Hylocomium splendens*, *Polytrichum* sp. and *Cladonia* sp. (see Gustafsson et al., 2019 for more details about the area). The area contains many small lakes (residence times mostly between 1-3 months), has a high peatland coverage (10-35%, Table 1, Fig. 1). The mineral soil consists of granitoid till, and is general thin where peatlands are not present. A wildfire started on July 31, 2014, and burned over 12 days covering an area of c. 13000 ha. The fire was low-intensity during the first days, but
spread rapidly when the wind speed increased and changed direction, and became a high-intensity stand replacing fire across all catchments investigated in our study. Due to the high intensity, fire fighting efforts were mostly restricted to protecting populated areas. Half of the burned area was salvaged logged during the first year  after the fire, while the other half was protected and left for natural regeneration.

The burned area consists of multiple catchments. We defined five major catchments in ArcGIS 10.3 (ESRI, Redlands, USA ) by using the Swedish elevation model (resolution 2×2 m and elevation accuracy of 0.5 m, Lantmäteriet 2015).When rain hits the surface it will run in the steepest slope direction which is determined in the elevation model. We delineated watersheds

by grouping the surfaces of the steepest slopes with the same direction. Two of these catchments are within the perimeter of the nature reserve with little salvage logging (Gärsjöbäcken and Vallsjöbäcken), while two are largely salvaged logged (Myckelmossen and Märrsjön, Table 1, Fig. 1). All catchments were close to completely burned and their outlets were placed just outside the burned area where water sampling were performed.

## 2.2 Stream water sampling and chemical analyses

Over three years post-fire we sampled outlet stream water from the five catchment outlets and near surface water from one lake (Märrsjön). One stream (Gärsjöbäcken) and the lake (Märrsjön) are included in the Swedish long-term monitoring program (Fölster et al., 2014), and therefore have a long period of pre-fire data (something which is relatively rare in studies of wildfire impacts). We extracted data for the sites from 2005 to the present day (Miljödata-MVM 2019). For all sites, post-fire stream sampling begun 2-3 weeks after the fire (c. 1 week after the first major post-fire rain event, >20mm) and continued with high temporal resolution during the first 4 months, and thereafter with longer intervals depending on season and stream. The lake was sampled slightly less frequently. The water sampling and subsequent water chemistry analysis were made according to the Swedish monitoring program using standard methods at the SWEDAC accredited geochemical laboratory at the Department of Aquatic Sciences and Assessment at the Swedish University of Agricultural Sciences. Metal ions were analysed with ICP-MS, $SO_4$ and $Cl$ were analysed by ion chromatography. $NH_4^+$ and $NO_2^- + NO_3^-$ were analysed with an autoanalyser. Total organic carbon (TOC) and total N (TN) were analysed by combustion on unfiltered water samples (Shimadzu TOC-VCPH with a TNM-1 module). By using unfiltered water samples we include organic material that was washed out by erosion. In these boreal ecosystems the composition of TOC is completely dominated by DOC (Laudon et al., 2004). DOC was measured together with TOC in one stream during the first year and these variables were highly correlated (r=0.98). We therefore use TOC as a proxy for DOC. Total organic N (TON) was calculated as: TON = TN - $(NH_4^+)$-N - $(NO_2^- + NO_3^-)$-N.

## 2.3 Pre-fire soil conditions and carbon and nitrogen losses

We estimated shrub, moss and organic soil C and N losses in three catchments, including the two largest ones (Vallsjöbäcken and Gärsjöbäcken). Our large-scale sampling in the three catchments was based on a systematic 300 by 300 m grid. At each intersection of the grid, a 314 $m^2$ circular plot (r=10 m) was established for sampling (i.e., 300 m between each sampling plot). Within the plot we established two perpendicular transects with 41 sampling positions (every metre and in the center). The high sampling density was chosen as burn severity is known to be extremely heterogeneous and spatial autocorrelation of organic soil depth is likely somewhere between 0.85 and 2.85 m (Kristensen et al., 2015). At each position, we registered the fire effect on the shrub layer (intact, only singed, only charcoaled stumps remaining, or totally consumed ). For non-peaty soils (>30 cm of organic matter), we measured depth of the remaining soil organic layer (to nearest half centimetre), and recorded whether the top layer (moss-lichen + $O_i$ horizon) had been consumed or not at each of the 41 positions within the plot. The ash layer (defined as "*the particulate residue remaining, or deposited on the ground,*

*from the burning of wildland fuels and consisting of mineral materials and charred organic components*", Bodi et al., 2014) was considered as remaining soil and was generally thin (0-0.5 cm). By including the ash layer in our measurements of remaining organic soil, we introduce additional uncertainty to our carbon loss estimates if C density is much different in this layer. To evaluate this effect we performed sensitivity analyses using ash C content, thickness and weight from another study

from the same burnt area (Perez-Izquierdo et al., 2020). The plot mean was used to estimate depth-of-burn (DOB) as the predicted organic soil layer depth (based on reference sampling outside the burnt area) subtracted by the remaining depth (e.g. Kelly et al., 2016; Turetsky et al., 2011). In peatlands, we measured DOB at each position by measuring the distance between the post- and pre-fire positioning of the organic layer. We reconstructed the pre-fire position using the positioning of adventive roots on the basal area of tree trunks, positioning of horizontal tree roots, and positioning of remnants of the

ground vegetation and peat mosses (for a detailed description of the methods see Kelly et al., 2016; Turetsky et al., 2011).

Carbon and nutrient losses during the fire were estimated for the organic soil layer and ground vegetation. To do this we needed to reconstruct the pre-fire organic soil thickness, bulk density, and nutrient content (C, N, S, K, Ca, P) of the organic soil layer, moss/lichen layer, and ground-layer cover of shrubs to calculate their biomass and ultimately their C and N

content. Using the same protocol as for the burnt plots, we collected data from 11 reference transects in the unburnt surroundings, amounting up to 60 plots (Fig. 1). These transects were placed from hilltops to valley with five to six plots per transect, covering young to old forests, similar to the area burnt. Peatlands were in included as we estimted depth of burn directly in these habitats.

For reference data on the organic soil layer, we sampled three to five soil cores (d=10 cm, depth = 5-30 cm depending on terrain) per plot and split them into a living moss/lichen section including the $O_i$ horizon, and a decomposed section (O-horizon consisting of horizons Oe and Oa). Each section was dried (65°C, until no further weight loss occurred ), weighted, mixed, and thereafter analysed for total element mass by Forest Research, UK. Elements were measured on a mass basis (g $kg^{-1}$) and converted into element bulk density (BD, g $cm^{-3}$). We used the DOB estimates and bulk densities values (moss-

lichen layer + $O_i$ and $O_{e+a}$ horizon) to calculate the soil C and N losses per area (DOB × BD). Unburned reference sites have often been used  as controls to estimate fire generated C and N losses (e.g. Kelly et al., 2016; Turetsky et al., 2011), and produce estimates similar to studies that used both pre- and post-fire measurements (Johnson et al., 2007). For peatlands we used published data on BD (5 cm depth interval, Granath et al., 2016) for boreal drained and undrained peatlands as the treed peatlands in the burned area in general are drained. Peat C content and N content were assumed to be 55% and 2%,

respectively (Minkkinen and Laine, 1998).

In our study we call these losses for direct losses (or emissions), meaning that they were predominantly lost from the soil and ground vegetation at the time of the fire. DOB data was collected within one year post-fire, and for uplands they were based

in the remaining organic soil layer. Hence, there is a possibility that we include other early losses (e.g. fluvial and respiration losses) in our upland direct emission estimates.

We estimated ground vegetation cover in the reference plots by recording the presence-absence of dwarf-shrubs at 41 positions within each plot. To convert cover to biomass we used species-specific relationships between cover and biomass for the major shrubs species (*Vaccinium myrtillus*, *V. vitis-idaea*, *Calluna vulgaris*, and *Rhododendron tomentosum*). In a second step, we scaled up C and N losses to catchment level by using the average losses for upland and peatland weighted by their coverage, respectively. Peatland cover was retrieved from the Swedish Geological Survey database (https://apps.sgu.se/kartvisare/kartvisare-torv.html).

C and N losses from standing trees were not estimated. It is very hard to make reliable quantifications of such losses (amount of fine branches and needles consumed) and the fuel amount varies with stand density and age. A typical pine stand in the burnt area may have 750 stems per hectare, a stem diameter between 15-20 cm and be 15-20 m high. This gives about 0.5 kg m$^{-2}$ C stored in living branches and needles, and 0.15 kg m$^{-2}$ C only in needles (calculated using allometric equations from Marklund 1988). Only 21% of the area experienced 100% crown damage, and about 50% between 50-100% damage (Gustafsson et al., 2019). Charred needles and fine branches were still visible in the burned pine crowns, indicating small losses from the trees and likely amounting up to a few percentages of the total C loss in forested areas.

## 2.4 Measuring $CO_2$ fluxes

Net ecosystem exchange (NEE) of $CO_2$ was measured by eddy covariance (EC) at two locations within the burned area (Fig. 1). Each EC system comprised a CSAT-3 sonic anemometer and an EC155 closed-path gas analyzer as an integrated system (CPEC200, Campbell Scientific, Logan, UT, USA). The sensors were mounted on a boom at the top of a 2 m tripod. Measurements were made at 10 Hz using a CR-3000 datalogger (Campbell Scientific, Logan, UT, USA). Meteorological measurements including air temperature, solar radiation, and soil moisture and temperature at 5 cm depth were recorded at the same location as 30 minute averages. Raw 10 Hz EC data were aggregated to calculate 30 minute average $CO_2$ fluxes, and overall fluxes were calculated according to the EUROFLUX methodology for error correction and gap-filling (Aubinet et al., 1999; Lee et al., 2004). In particular, detrending was applied using a digital, recursive filter with a time constant of 2000s, and the covariance matrix was aligned with the mean wind vector by a two-fold coordinate rotation on a half-hourly basis. Data analysis was done using R (R Development Core Team, 2016) and the R package *Openair* (Carslaw and Ropkins, 2012). The EC systems were installed in april 2015 due to limitations to access the burnt area and  $CO_2$ fluxes prior to that date (autumn-winter) were modelled. For a more detailed description of the data processing and gap-filling techniques used, see Hadden and Grelle (2017).

## 2.5 Element budget calculations

To make approximate element budgets we combined estimates of pools and fluxes in the system. Our aim was not to make a complete budget, but rather to contrast immediate changes in stocks (assumed to be direct gaseous emissions for N and C) during the fire and subsequent (leached out or net ecosystem $CO_2$ exchange) losses from the ecosystem. Pre-fire element pools were derived from reference sites and emissions were estimated from DOB (see text above). This was done for the two major catchments (Gärsjöbäcken, Vallsjöbäcken) for which we had DOB measurements. Fluvial transported material was calculated based on stream flow and water element concentrations. Flow data were based on S-HYPE (Strömqvist et al., 2012), the national application of the HYPE hydrological model (Lindström et al., 2010). HYPE is a process-based daily time-stepping catchment model. In a HYPE model application the modelled domain is divided into sub-basins with unique distributions of hydrological response units (HRUs). These HRUs are typically a combination of specific land-uses and soil types. The soil profile of each HRU may contain up to three soil layers. Runoff of water from the soil layers including overland flow are simulated and summed for each HRU. and routed through the network of rivers and lakes in the model. The Vallsjöbäcken catchment was extracted from the national model application and calibrated against local pre-fire and post-fire streamflow data using an automatic calibration routine. Pre-fire data were obtained from a stationary streamflow gauging station in operation until the early 2000s. Post-fire streamflow time-series were derived from data from installed pressure transducers and a rating curve developed from the recorded water level and flow measurements. The post-fire model was validated against streamflow data derived from the transducer installed in Gärsjöbäcken. Using this model we also extracted daily estimates of the average residence time water in the drainage network upstream of the sampling point.

Element mass flow was calculated as daily flow × element concentration. As element concentration was not measured daily we used predicted values from a model that made linear predictions between time points. This approach (period-weighted) was chosen over a model based on flow - concentration relationships because such relationships were weak in our data, indicating that non-hydrological factors dominated observed temporal variations (see Results). Our approach is recommended by Aulenbach et al. (2016) when there is a weak concentration - discharge relationship and the load estimate error should not be larger than 5-10% (Aulenbach et al., 2016). Element outflow was aggregated over time and we present values for 3 years pre-fire (for Gärsjöbäcken catchment, the long-term monitoring site), and for 3 years post-fire (Gärsjöbäcken and Vallsjöbäcken).

## 2.6 Element decay curves and pH modelling

For solutes that showed a single 'pulse' response to the fire (Cl⁻, $Ca^{2+}$, $Mg^{2+}$, K⁺, $SO_4^{-2}$, $NH_4^+$, TN), we fitted exponential decay curves to observed concentrations, in order to derive a set of diagnostic parameters describing the magnitude of fire response and rate of recovery to pre-fire baseline conditions. This procedure was undertaken at the four streams with sufficient data to support curve-fitting: Myckelmossbäcken, Ladängsbäcken, Gärsjöbäcken and Vallsjöbäcken. Based on an initial assessment

of the data, it was apparent that some solutes did not follow a simple (single) exponential decay curve, whilst in all cases solute concentrations converged on a non-zero baseline concentration towards the end of the measurement period. Therefore we conceptualised the change in solute concentrations according to the equation (1):

$$C_t = C_{baseline} + C_{fast} \times 0.5^{(t/t\frac{1}{2}fast)} + C_{slow} \times 0.5^{(t/t\frac{1}{2}slow)} \tag{1}$$

Where Ct is represents solute concentration at time t, $C_{baseline}$ is the average concentration of a solute in the absence of fire effects, and $C_{fast}$ and $C_{slow}$ are the maximum post-fire concentrations of two exponentially declining pools, one ), with associated half-lives t½fast and t½slow.

For each solute at each site, we fitted  non-linear decay curves (equation 1). First, we located the time of peak measured
concentration at each site (which was not necessarily the same at all sites and nor the first measurement post-fire) as time zero. Next, we estimated $C_{fast}$, $C_{slow}$, t½fast and t½slow to each solute time series by using a Bayesian approach in the R package *brms* ver 2.10 (Bürkner, 2017). To regularise estimation we used weakly informative (proper) priors based on expected values: mean and sd 10 for $C_{baseline}$ and  t½fast , mean 100 and sd 25 for  $C_{fast}$ and $C_{slow}$ and  t½slow). Least-square estimation gave similar results but was sensitive to starting values for each model. The fast-pool was tested by examining if the 95%
credible intervals of the fast-pool parameters included zero. With the fitted models we defined pool half-lives; the amount and relative proportion of peak measured concentrations associated with baseline, fast- and slow-decay pools; and the ratio of peak to baseline concentrations for each site and solute combination.

We modeled pH and charge of organic anions ($RCOO^-$) following the approach by Köhler (2000) which is based on TOC,
alkalinity and $pCO_2 = 2.8$ using the CBALK approach. Charge balance with respect to buffering capacity and organic anions is achieved through iteration until a charge balance criteria of positive and negative charges ($< 0.1$ µeq $l^{-1}$) is met. pH measurements were taken coincident with the water samples to validate this model.

**2.7 Leaf Area Index**

To examine post-fire plant regrowth, we extracted remotely sensed Leaf Area Index (LAI) at peak growing season (June 15 -
July 28) for 2014 (before fire) to 2019. We downloaded MODIS LAI data (product: MCD15A2H) with a 500 m pixel size and 8-day averages (Myneni et al., 2015). We filtered out 'bad' pixels using the quality layers (e.g. pixels with clouds and high aerosol content). Pixels covering more than 25% water were also removed from further calculations. Finally, we extracted the mean values for each catchment and year. MODIS data were downloaded in R using the MODISTools package (ver 1.1.1, Tuck et al., 2014) and calculations were performed with the raster package (ver. 3.0-7, Hijmans et al., 2019).

## 3 Results

### 3.1 Element losses and C fluxes

C and N losses from the soil and ground vegetation during the fire (assumed to be emission) were similar in the two focus catchments (Table 2). In forest (non-peaty) soils and ground vegetation, most of the C and N losses were from the O-horizon while the contribution of the shrub vegetation was negligible (1-2%). The moss/lichen layer (pre-fire thickness 39±15 mm, mean±SE) was, with rare exceptions, completely consumed by the fire. On average, 16 mm of organic soil remained after the fire (compared to an estimated 98±53 mm pre-fire) and the organic soil C and N stock had been drastically reduced (-88%).

Fluvial element transport was controlled mainly by element concentration, as we found no evidence that element concentration was a function of stream flow. For the two catchments, flow explained at the most (for K at Vallsjöbäcken) 17% of the variation in element concentration, followed by $SO_4^{2-}$ with 10 to 11% explained variation for the two catchments (Fig. S1). In the Gärsjöbäcken catchment that had pre-fire data, the stream flow - element concentration relationship was equally weak the years before the fire ($R^2 < 20\%$). Furthermore, a pre- vs. post-fire comparison showed that fluvial losses increased drastically for all elements and were around 5 times higher during the first year, except for S that was 26 times higher (Table 2). In the third year post-fire, S and P still showed higher values than before the fire, whilst Ca and K had returned to pre-fire levels. For Vallsjöbäcken catchment, fluvial losses were overall lower than for Gärsjöbäcken but the temporal trend was almost identical. Discharge was substantially higher the first year (50-60%) in the two catchments, but thereafter similar to the pre-fire values.

Carbon fluxes were similar at the two sites and, on average, these two sites lost 158 g C m$^{-2}$ the first year, and in total ~440 g m$^{-2}$ (426 and 456 g m$^{-2}$) over three years (Fig. 2ab). This is about 10% of the C lost in the fire. Merging all C losses and fluxes over the first three years, we estimated the total C loss to be circa 4900 g m$^{-2}$ in the two catchments. There was a net C loss for all months except for a few summer months close to three years post-fire. This trend towards a net carbon uptake was mirrored in the large-scale vegetation regrowth data. Regrowth (here as LAI) occurred in similar rate among the burnt areas of the catchments (Fig. 2c). Ladängsbäcken, where 28% of the catchment area did not burn, showed a weaker response when LAI was estimated for the whole catchment (lowest value 1.84).

### 3.2 Water quality

Nitrate and ammonium concentrations increased rapidly post-fire and ammonium quickly decreased and stabilized within a 12 months in all catchments (Fig. 3). Nitrate, however, continued to show spring pulses. Soluble P also increased in streams but the magnitude varied and there are indications of winter-spring pulses. $SO_4^{2-}$, $Ca^{2+}$ and $K^+$ concentration followed the same pattern as ammonium and had stabilized after a year, except for K that returned at a slower pace. Fire had a marginal

effect on pH and TOC in streams (Fig. 3, Suppl Material Fig. S2, S3). A short acidification pulse (0.5-1 pH unit) occurred during the first few months but then pH slowly increased over time. Analyses of ions indicate that the pH was relatively stable after the fire because increases in acidity caused by $SO_4^{2-}$ were counterbalanced by organic acids and an increase in base cations ($Ca^{2+}$, $Mg^{2+}$, $Na^+$ and $K^+$)(Suppl Material Fig. S2-S4). The pH modelling exercise resulted in a median difference between measured and modeled pH in this data set of 0.19 pH unit. The large majority (> 90%) of the measured pH could be modelled within 0.5 pH units, which is in line with earlier similar studies (Fig. S5).

Examining the long-trends revealed that $PO_4^{3-}$, $SO_4^{2-}$ and $K^+$ concentrations had not completely returned to pre-fire values after three years, either in the lake (not for P) or the stream (Fig. 4). Moreover, the lake data did not show a strong response to the fire, although the stream and lake did not differ much in the pre-fire values and the whole lake catchment burnt severely.

### 3.3 Decay curves

Solute peaks were identified circa 1-3 months post-fire, with the two larger focus catchments (Gärsjöbäcken and Vallsjöbäcken) peaking later than the smaller catchments. Fitted solute decay curves are shown for the most intensively sampled site, Gärsjöbäcken, in Figure 5. Summary data from the curve fitting for all four streams are shown in Table 3. For three of the four streams, the inclusion of a fast-decaying pool improved the model fits for most solutes, whereas at the strongly lake-influenced Vallsjöbäcken (flows through the largest lake), only a slow-decay pool was required to reproduce observations. Where present, the fast-decay pool contributed between 30% and 75% of post-fire peak concentrations, depending on site and solute, and typically had a $t\frac{1}{2}_{fast}$ of 4-20 days. The contribution of the slow-decay pool varied very widely, from < 10% to > 90% of peak concentrations, with a $t\frac{1}{2}_{slow}$ of 50-200 days.

We observed consistent differences in ratio of peak/baseline ratios as a function of both site and solute. In relation to site, ratios for all solutes followed the general pattern Myckelmossbäcken > Ladängsbäcken $\simeq$ Gärsjöbäcken > Vallsjöbäcken. This sequence appears to be inversely related to the relative influence of lakes (% lake cover of the catchment and distance to large water body, Table 1 and Figure 1) in the catchment upstream of the sampling point. In relation to solute, peak/baseline ratios typically followed the sequence $NH_4^+$ > $SO_4^{2-}$ > $K^+$ > TN $\geq$ $Ca^{2+}$ $\simeq$ $Mg^{2+}$ $\simeq$ $Cl^-$. This sequence was largely replicated in the half-life data, with solutes with high peak/baseline ratios also having the shortest $t\frac{1}{2}$ values. On the other hand, we found very little evidence to suggest that $t\frac{1}{2}$ values varied consistently between the four streams.

### 3.4 Sensitivity analyses

Hydrological losses could have been underestimated if a flush of nutrients occurred in the first three weeks after the fire, prior to the start of sampling. However, the amount of precipitation was not very large in this period, so the export flux of

water was low, thus solute concentration would have needed to be extremely high to generate a large solute export during this period. We consider this highly unlikely, because several catchments showed solute concentration peaks a few weeks after our first sampling point, indicating that flushing (at a catchment scale) often was delayed due to buffering in the system. A sensitivity analysis for the Gärsjöbäcken catchment, assuming that the carbon and nutrient concentrations one week after the fire were double the values measured as the first time point, showed that the impact on the annual budget in this extreme example would nevertheless be small,resulting in an underestimation of circa 0.5% for carbon and 1% for nitrogen.

Treating the thin ash layer as unburned organic soil likely led to some underestimation in our carbon loss estimates, due to the lower C density in ash compared to the organic soil. Using a (high) estimated ash thickness of 1 cm, a C content between 20 to 25%, and a wide observed ash weight (ash data from Pérez-Izquierdo et al., 2021), we calculate that treating the ash layer as unburnt organic soil could have resulted in an underestimate of the average calculated carbon loss in the range 0.01 to 1% (2 to 45 g C m$^{-2}$).

We did not include losses from downed wood in our C losses as this is a small component in this managed landscape. The burnt area had before the fire around 4 m$^3$ per hectare of downed wood (Jonsson et al. 2016). Assuming a stem density of 412 kg m$^{-3}$ for Scots pine (Repola 2006), and 50% carbon content, the maximum loss from downed wood is on average about 80 g C per m$^2$ (or around 1.5% of our calculated total C loss). This maximum value is likely an overestimation as downed wood was rarely completely consumed by the fire.

In combination, we estimate that these potential omissions in our budget calculations could have led to an underestimate of soil and forest floor total C loss of less than 3%. Effects on budget calculations for other elements are likely smaller.

## 4 Discussion

### 4.1 Element balances

Our study shows that fire-related C and N losses resulting from a boreal wildfire were dominated by losses of the C stocks in soil O horizons, and we ascribe these losses to direct emissions during the fire (see Fig. 6 for a summary on C). Post-fire fluvial C and N losses were almost negligible compared to the deep burns in forest and peatland soils. This illustrates the importance of correctly estimating how much organic matter was consumed in the fire compared to other losses for calculating C and N budgets. The amount of C lost in the fire is around 200-1000 times higher than reported annual riverine export from boreal catchments (5-8 g m$^{-2}$ y$^{-1}$; Laudon et al., 2004). We did not observe increased fluvial C losses during the first year after the fire, despite the increased discharge caused by a thinner organic soil layer that decreases catchment water storage in combination of ceased plant water use. Some earlier work has suggested that fluvial dissolved C loss increases post-fire, for both wildfires (Emelko et al., 2011; McEachern et al., 2000; Minshall et al., 2001) and prescribed fires

(Mitchell and McDonald, 1995; Yallop et al., 2010). However, our results for TOC (considered to largely comprise DOC as discussed above) are more in line with more recent research that has found little or no effect of fire on DOC export (Betts and Jones, 2009; Burd et al., 2018; Evans et al., 2017).

Net ecosystem exchange (NEE) of $CO_2$ over the first 3 years post-fire indicated larger post-fire C loss than hydrologically exported C, but it still only comprised 10% of the direct combustion emissions. It should be noted that our estimates of direct emissions may include early respiration and leaching losses, but given the severity of the fire with deep burns and large losses we consider that combustion losses comprised the large majority of this loss. In addition, we likely underestimated direct C emissions as we did not include downed wood or biomass losses from living trees. With no vegetation, it is no
surprise that the system acted as a C source immediately after fire, and the observed release of $CO_2$ can mainly be ascribed to heterotrophic soil respiration, and to a smaller extent from dead needles and woody biomass. Compared to undisturbed systems, heterotrophic respiration actually seems to decrease after fire (reviewed in Amiro et al., 2003), partly due to the formation of inert carbon, i.e. pyrogenic carbon that may stabilise the remaining organic carbon (Jones et al., 2019). About three years post-fire, summer NEE showed for the first time net C uptake. It is likely that the overall pattern was similar
across the whole burn, because we observed a rapid increase in LAI in all catchments. Flux data from boreal North America have also shown summer net C uptake two years post-fire, but it may take 10 years until the system is a sink on an annual basis (Amiro et al., 2010; Amiro et al., 2003; Goulden et al., 2011; Kashian et al., 2013). Specifically, an eddy covariance study in boreal Canada estimated the Net Ecosystem Production one and two years post-fire and reported C losses of 192 and 93 (g C $m^{-2}$ $yr^{-1}$) respectively (Goulden et al., 2011). These values are similar to our two sites (155 to 165 g C $m^{-2}$ $yr^{-1}$ over
two years), but further research is needed to establish if such values are typical of boreal uplands post-fire. However, despite our effort to track carbon flows in the system, we still had to model flux values for the first fall-winter period, and combustion losses were inferred by using unburned reference plots. Hence, our estimates are associated with uncertainty that needs to be considered when upscaling these results.

In contrast to C, we observed a dramatic increase in hydrological N loss that was largely driven by higher concentrations in the streams. The amount of dissolved N lost over the first years (almost 1 g N $m^{-2}$) may be small compared to the direct combustion losses (<1%), but this is available N, whereas much of the N lost in the fire is N would have derived from stable organic matter forms that were not readily available for the plants (Smith et al., 2011; Tamm, 1991). Our annual estimates of fluvial N losses are similar to those reported for a mixed coniferous forest (Nevada, US, Johnson et al., 2007) and for peaty
heathland (Northern Ireland, UK, Evans et al., 2017), but 100 times greater losses than has been reported for a Mediterranean shrubland (Dannenmann et al., 2018). Our estimates of direct N losses are at the higher end of reported values for temperate/boreal coniferous forests 30 to 90 g $m^{-2}$ (Brais et al., 2000; Grier, 1975; Johnson et al., 2007), but in contrast to previous studies, we included N losses from drained peatlands that probably resulted in higher total losses. In addition, both

other estimates and our own do not include post-fire gaseous N emission that during the first post-fire year has been shown to comprise 10-15% of the of the direct fire combustion losses in shrubland systems (Dannenmann et al., 2018).

Despite these large N losses, there is little evidence that either direct or fluvial N losses are relevant for post-fire productivity at a catchment scale. Recently, a study by Turner et al. (2019) showed a remarkably rapid post-fire (4 years) build-up of soil N, and little evidence that the N loss had an long-term impact on productivity. It is hypothesised that post-fire plant communities, if quickly established, can retain N before it is lost hydrologically (Smithwick et al., 2009). At our study site, vegetation established after 2 years, but most of the soluble N had already been lost at that time. Hence, plants must utilize newly mineralised N, or acquire their N through microbes (e.g. via N-fixation). In fact, it is unknown how plants can acquire large amounts of N post-fire and how the N pool builds up quicker than estimated N-fixation rates (Turner et al., 2019). Our N losses (ca. 100 g m$^{-2}$), for example, correspond to more than 150 years of N input from fixation and deposition (based on 0.6 g m$^{-2}$ yr$^{-1}$ N input) (Brais et al., 2000; Zackrisson et al., 2004). Clearly, fire is a key driver of the global N cycle. Future studies should focus on elucidating the mechanisms behind post-fire N build-up in the boreal biome to better capture this dynamic in ecosystem models.

Few studies have quantified other fire-related nutrient losses such as S, P, K and Mg. The integrated hydrological mass export during the first year after the fire corresponds to around 5 years (P, K and Mg) and 26 years (S) of pre-fire element export (Table 2). Hence, on a longer time-scale, these losses seem unlikely to affect the productivity of the system, although they could influence short-term availability for uptake by the biota, as well as soil acidity, in these relatively base-poor ecosystems. Instead, our study indicates that soil and biomass retention capacity for base cations was fast and efficient in this fire impacted boreal ecosystem. Shorter fire-intervals might therefore have a limited impact on base cation budgets, although it is clear that they will fundamentally alter C and N budgets due to loss of slow-forming organic soil.

**4.2 Water quality and decay curves**

The fire had generally a strong short-term impact on the water quality with large short-term variations of both base cations and acid anions over time. Hence,  our study highlights the importance of frequent sampling soon after the fire to accurately capture the post-fire dynamics in water chemistry. At these peatland-rich sites, pH remained fairly stable despite the great fluctuations in mineral anions ($SO_4^{-2}$, $NO_3^-$; Fig. 3). A significant pH drop only occurred at the peatland dominated site Myckelmossbäcken where TOC was initially suppressed just after the fire (Fig. 3, Fig. S3). At all other sites organic anion concentrations were above 100 µeq l$^{-1}$ (Fig. S4) and buffered pH against any potential charge imbalance of sulfate and base cations. TOC is mainly released from riparian peatlands in boreal catchments (Ledesma et al., 2015) and it is possible that an intact (less burned) riparian zone through its TOC release can buffer and thus prevent a large pH drop from occurring after fire.

Sustained elevated levels of reactive phosphorus have been reported for other boreal wildfires, and our relative increase are similar to studies examining phosphorus concentration up to five years post-fire (Hauer and Spencer, 1998; Silins et al., 2014). The P and N enrichment likely caused higher algal productivity in streams, which can generate effects at higher trophic levels (Silins et al., 2014), but this was not monitored in our study.

The analysis of decay curves suggests that there are two distinct sources of solute flushing to the stream. A single-exponential model was unable to reproduce both the rapid initial decline and the longer-term decrease, whereas a two-pool model generally gave a good fit. Moreover, a two-pool model is mechanistically interpretable. The first 'fast-decay' pool is associated with the immediate post-fire period, typically made a significant contribution to peak solute concentrations, and was observed for most solutes in three of the four streams, with the exception of the strongly lake-influenced Vallsjöbäcken. The rapidity with which this peak dissipated, with half-lives between 4 to 25 days, suggests that it reflects the instantaneous mobilisation of solutes due to pyrolysis of biomass and soil organic matter, followed by hydrologically-controlled flushing into the drainage network. The fine ash that formed is most probably very soluble and may be leached out fast with rainwater (Grier 1975). The second, 'slow-decay' pool contributed variably to post-fire peak concentrations, but affected water chemistry for a period of years, with half-lives typically in the order of 75-175 days. The consistent differences in $t^{1/2}_{slow}$ between solutes, coupled with the absence of clear variability in $t^{1/2}_{slow}$ between streams, leads us to conclude that this pool is largely determined by biogeochemical processes occurring after the fire. This fits with the observed heterotrophic respiration in our NEE data, and suggests gradual leaching of solutes from ash and the breakdown and dissolution of dead organic matter. The relative contribution of the two pools of element leaching is likely determined by burn severity, where a more severe burn would increase the size of the fast pool by consuming more of the organic matter, leaving the inorganics (K, Ca, $NH_4^+$ etc) available for rapid leaching. This would also suggest that more severe fires result in a smaller 'slow' pool, because there is  less organic matter left to decompose.

Differences in peak/baseline ratios and $t^{1/2}_{slow}$ between solutes appear to reflect their source within the ecosystem; N and K are largely present in non-woody biomass including microbes, leaves and fine roots, and are therefore likely to be released relatively quickly. In particular, $NH_4^+$ is the initial product of organic matter mineralisation, and the very large and fast-declining peaks observed in this solute (e.g. compared to either $NO_3^-$ or TN) suggest that the supply of $NH_4^+$ due to organic matter pyrolysis and mineralisation immediately following the fire overwhelmed abiotic and biotic retention mechanisms, as well as terrestrial and aquatic nitrification capacity. This short-lived $NH_4^+$ pulse, together with more sustained leaching of $NO_3^-$ in the years after the fire, is consistent with previous studies of wildfire impacts (e.g. Wan et al., 2001) and with other studies of N cycle responses to major ecosystem disturbances, such as bark beetle attacks (Kopáček et al., 2018). The mechanisms behind such similar responses to different disturbances are likely less plant uptake and increased N mineralisation. In contrast to N solutes, the divalent base cations are more structurally bound within biomass pools, and strongly retained on soil cation exchange sites, and therefore released more gradually via organic matter mineralisation,

especially in presence of pyrogenic organic matter. The slow release of Cl also suggests release from decaying organic matter, consistent with previous studies suggesting that large amounts of Cl is biotically cycled within Northern forest ecosystems (Bastviken et al., 2006). The source of $SO_4^{-2}$ leaching may be somewhat different, because the largest pools of S in our study catchments are believed to be sulphides and organic S compounds held under anaerobic conditions in wetlands (Schiff et al., 2005). Thus the largest peaks in $SO_4^{-2}$ were recorded in the peat-influenced Myckelmossbäcken, associated with the combustion of a considerable depth of peat.

In two of the investigated stream catchments most of the forest stands were salvage logged during the first year after the fire. Interestingly, we did not observe any clear or consistent differences on water quality between salvage logged and non-salvage logged catchments over the study period. A study by Silins et al. (2014), possibly the only study that has made this comparison for boreal a catchments, found larger increases of stream P concentration in salvage logged catchments. However, this was in an area with extreme topography (Rocky Mountains) where mechanical damage led to increased erosion. In our lower-relief study area, evidence of large-scale soil disturbance during salvage logging was not observed. We did not investigate post-fire $CO_2$ fluxes in logged areas, but previous studies have not found clear evidence of increased soil C losses compared to unlogged areas (Marañón-Jiménez et al., 2011; Kishchuk et al., 2016). To better investigate if post-fire salvage logging has an ecologically important effect on water quality in boreal Europe more catchments and longer time series are needed.

**5 Conclusions**

Our study provides a unique integrated quantification of the impact of wildfires on boreal forest biogeochemistry (e.g. Fig. 6). Overall, hydrological export of nutrients was fairly short-lived (1-2 yr), and was caused mainly by higher ion concentrations, and not by increased discharge. For some major elements with gaseous loss pathways, notably C and N, fluvial losses were small compared to the direct emission; in fact, no increase in aqueous C export was observed and fluvial losses of N and C can be considered minor compared to combustion losses for boreal catchment budgets during a fire. For elements that showed elevated exports (N, P, S, Mg, K), the first year post-fire was equivalent to circa 5 years (26 for S) of exports in unburned systems. Base cation fluxes three years post-fire were similar to pre-fire conditions except for K that remained elevated much longer, suggesting slower release and weaker retention of this element. Our decay curves and comparable pre- and post-fire fluxes indicate that the boreal forest ecosystem has re-established a similar steady-state of deposition, weathering and export. It will be interesting to revisit these catchments in a few years to study whether the element uptake of the growing trees will lead to lower stream water export. Vegetation regrowth was rapid and likely contributed to decreased leaching of nutrients while initiating C sequestration of the system. Three years post-fire, there was a clear net ecosystem C uptake during the summer, suggesting that fire-induced C losses had largely concluded, and that the ecosystem will likely become a net $CO_2$ sink in future years as the forest regrows. However, given the magnitude of C loss

from combustion of the organic soil it will likely take decades or even centuries for overall ecosystem C stocks to recover. If fire frequency increases across boreal forest ecosystems these forests can become net long-term sources of $CO_2$ to the atmosphere, reversing their current function as carbon sinks.

## 6 Data and code availability

5 Data and R code are currently available for review at https://github.com/ggranath/FireMassFlows and will be available at zendo with DOI if manuscript is accepted.

## 7  Supplementary material

Figure S1 to S5.

## 8 Author contribution

10 Overarching research objectives were formulated by GG, SJK and CDE. GG and JStrengbom designed the soil and vegetation sampling scheme, collected data on depth of burn, and calculated carbon losses during the fire. SJK and JF designed and coordinated sampling and lab analyses of water chemistry, and did the pH modelling. CDE developed the concept of decay curves. JStrömqvist calculated stream flow and water balance. AG established and maintained the eddy covariance towers and calculated carbon exchange based on their data. GG performed the nutrient balance analyses, LAI

15 analyses, and were responsible for the overall data analyses, GIS work, and graphical presentation. GG wrote the first draft with input from CDE and SJK. All authors read and commented on the manuscript and approved the final version.

## 9 Competing interests

The authors declare that they have no conflict of interest.

## 10 Acknowledgement

20 We thank Anna Landahl and Jessica for helping out collecting the data. The study was supported by the Swedish agency for Marine and Water Management (grant 1:12, Skogsbranden i Västmanland 2014: Utvärdering av effekter på vattenkvalitet och vattenlevande organismer i och runt brandområdet), and by FORMAS (Swedish Research Council for sustainable development) (grants 2014-01850 and 2014-01869). C.D. Evans contributed to the study as part of a King Carl XVI Gustaf visiting professorship at SLU.

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

| Catchment | lon/lat | Area (ha) | Lakes (%) | Upland forest (%) | Forested peatlands (%) | Open peatlands (%) | Prop. burned (%) | Prop. logged (%) | Mean outflow age (yr) |
|---|---|---|---|---|---|---|---|---|---|
| Gärsjöbäcken* | 16.22887 59.92148 | 2170 | 2 | 66 | 17 | 15 | 100 | 2-3 | 0.25 |
| Vallsjöbäcken | 16.091216 59.882230 | 1830 | 3 | 79 | 13 | 5 | 96 | 5-10 | 0.29 |
| Ladängsbäcken | 16.17334 59.81866 | 1440 | 0 | 72 | 16 | 12 | 72 | 0-1 | 0.007 |
| Myckelmossbäcken | 16.27152 59.89688 | 930 | 0 | 73 | 15 | 12 | 96 | 40-55 | 0.007 |
| Märrsjöbäcken | 16.04151 59.93975 | 374 | 14 | 73 | 9 | 4 | 86 | 40-60 | 1.47 |
| Märrsjön (lake)* | 16.05693 59.94867 | 233 | 23 | 63 | 9 | 5 | 100 | 50-63 | |

**Table 1. Overview of the burnt catchments, their land characteristics, and annual mean outflow water age (2014 aug - 2015 july). The last catchment (Märrsjön) is a lake catchment. Proportion logged is based on estimated salvage logged area during the first year after the fire. Longitude and latitude (WGS84) indicate the sampling location. See also the map in figure 1. Long-term monitoring catchments are indicated with an asterisk (\*).**

| Element | Catchment | Pre-fire exported (g m$^{-2}$ yr$^{-1}$) | Emission (g m$^{-2}$)* | Post-fire exported, Aug-July (g m$^{-2}$) | | | Fire-effect on exported masses or water (ratio)*** | Pre-fire storage (forest O-horizon)** (g m$^{-2}$) | Post-fire storage (O-horizon)** (g m$^{-2}$) |
|---|---|---|---|---|---|---|---|---|---|
| | | 2010-13 | During fire | 2014-15 | 2015-16 | 2016-17 | | | |
| C | Gärsjöb. | 7.505 | 4 204 | 8.162 | 6.580 | 4.381 | 1.1, 0.9, 0.6 | 4 987 | 568 |
| | Vallsjöb. | | 4 560 | 6.383 | 4.113 | 2.899 | | 4 987 | 412 |
| N | Gärsjöb. | 0.149 | 98 | 0.836 | 0.244 | 0.136 | 5.6, 1.6, 0.9 | 138 | 21 |
| | Vallsjöb. | | 114 | 0.398 | 0.187 | 0.124 | | 138 | 15 |
| P | Gärsjöb. | 0.00389 | | 0.0196 | 0.0098 | 0.0055 | 5.0, 2.5, 1.4 | 9.8 | |
| | Vallsjöb. | | | 0.011 | 0.0066 | 0.0044 | | 9.8 | |
| S | Gärsjöb. | 0.040 | | 1.042 | 0.158 | 0.072 | 26.0, 3.9, 1.8 | 10 | |
| | Vallsjöb. | | | 0.623 | 0.134 | 0.079 | | 10 | |
| Ca | Gärsjöb. | 0.637 | | 2.921 | 1.000 | 0.672 | 4.6, 1.6, 1.1 | 50 | |
| | Vallsjöb. | | | 2.168 | 0.914 | 0.690 | | 50 | |
| K | Gärsjöb. | 0.278 | | 1.304 | 0.321 | 0.144 | 4.7, 1.2, 0.5 | 34 | |
| | Vallsjöb. | | | 0.971 | 0.266 | 0.126 | | 34 | |
| | | | | | | | | | |
| | | (m) | | (m) | (m) | (m) | | | |
| water | Gärsjöb. | 0.28 | | 0.42 | 0.30 | 0.21 | 1.5, 1.1,0.75 | | |
| | Vallsjöb. | 0.24 | | 0.38 | 0.28 | 0.19 | 1.6, 1.2, 0.8 | | |

*average loss over peatlands and forests, **peatlands excluded, *** for 2014, 2015 and 2016

**Table 2. Emissions during the fire and hydrologically exported masses for major elements. Pre-fire data is estimated for one catchment (Gärsjöbäcken catchment) as a 4 year average (2010-2013). Post-fire exported masses are calculated over 12 months periods after the fire (August-July) for three years (2014-2017) with an expected error of 5-10% (Aulenbach et al., 2016). Amounts are calculated from total N, total P, $SO_4^{2-}$-S, Ca, K and total organic C (TOC in streams). Emission is estimated losses to the air during the fire, excluding trees (dead and living). Pre-, and post-fire storage are only for the forest habitat (excluding peatlands). Water is discharge per year, normalised for catchment area, and expressed in metres.**

| Determinand | Baseline | | Fast pool | | | Slow pool | | | Peak/baseline |
|---|---|---|---|---|---|---|---|---|---|
| | μmol l$^{-1}$ | % | μmol l$^{-1}$ | % | T½$_{fast}$ | μmol l$^{-1}$ | % | T½$_{slow}$ | |
| **Myckelmossbäcken** | | | | | | | | | |
| Cl | 58 | 21 | 88 | 31 | 15 | 137 | 48 | 172 | 4 |
| Ca | 31 | 10 | 141 | 45 | 24 | 142 | 45 | 110 | 9 |
| Mg | 27 | 11 | 122 | 52 | 17 | 86 | 37 | 81 | 8 |
| K | 15 | 11 | 18† | 13† | 10† | 103 | 76 | 140 | 8 |
| SO4 | 13 | 5 | 150 | 52 | 19 | 123 | 43 | 74 | 21 |
| NH4 | 1 | 1 | 100 | 56 | 11 | 79 | 44 | 80 | 138 |
| TN | 39 | 16 | 105 | 44 | 21 | 97 | 40 | 107 | 5 |
| **Ladängsbäcken** | | | | | | | | | |
| Cl | 45 | 22 | 72 | 35 | 15 | 88 | 43 | 120 | 4 |
| Ca | 31 | 12 | 118 | 47 | 13 | 103 | 41 | 101 | 7 |
| Mg | 31 | 17 | 94 | 53 | 10 | 53 | 30 | 80 | 5 |
| K | 14 | 14 | 9† | 9† | 13† | 76 | 77 | 127 | 6 |
| SO4 | 10 | 5 | 106 | 56 | 10 | 73 | 39 | 84 | 19 |
| NH4 | 1 | 1 | 95 | 60 | 8 | 64 | 40 | 90 | 183 |
| TN | 36 | 14 | 121 | 48 | 12 | 95 | 38 | 98 | 6 |
| **Gärsjöbäcken** | | | | | | | | | |
| Cl | 47 | 25 | 27 | 14 | 11 | 116 | 61 | 168 | 3 |
| Ca | 64 | 19 | 136 | 40 | 22 | 143 | 42 | 141 | 4 |
| Mg | 29 | 18 | 76 | 45 | 12 | 62 | 37 | 129 | 5 |
| K | 16 | 11 | 35 | 25 | 9 | 89 | 64 | 152 | 8 |
| SO4 | 10 | 5 | 73 | 40 | 7 | 98 | 54 | 114 | 17 |
| NH4 | 1 | 0 | 91 | 42 | 4 | 126 | 58 | 94 | 247 |
| TN | 44 | 15 | 130 | 44 | 7 | 119 | 41 | 153 | 6 |
| **Vallsjöbäcken** | | | | | | | | | |
| Cl | 55 | 34 | 34 | 21 | 12 | 73 | 45 | 149 | 2 |
| Ca | 74 | 34 | 51 | 23 | 16 | 94 | 43 | 138 | 2 |
| Mg | 38 | 43 | 11† | 12† | 14† | 41 | 46 | 108 | 1 |
| K | 15 | 16 | 5† | 5† | 13† | 78 | 79 | 135 | 5 |
| SO4 | 11 | 12 | 11† | 12† | 13† | 72 | 76 | 110 | 7 |
| NH4 | 2 | 4 | 7† | 13† | 16† | 44 | 83 | 82 | 25 |
| TN | 44 | 39 | 21† | 19† | 13† | 47 | 42 | 112 | 2 |

**Table 3. Decay curve modelling. Fitted initial/baseline pool concentrations, percentages, pool half-lives and peak/baseline ratios for solutes exhibiting a single post-fire concentration peak in the four studied streams. If the addition of a fast-decay solute pool did not improve the model fit, these cells are left blank.**

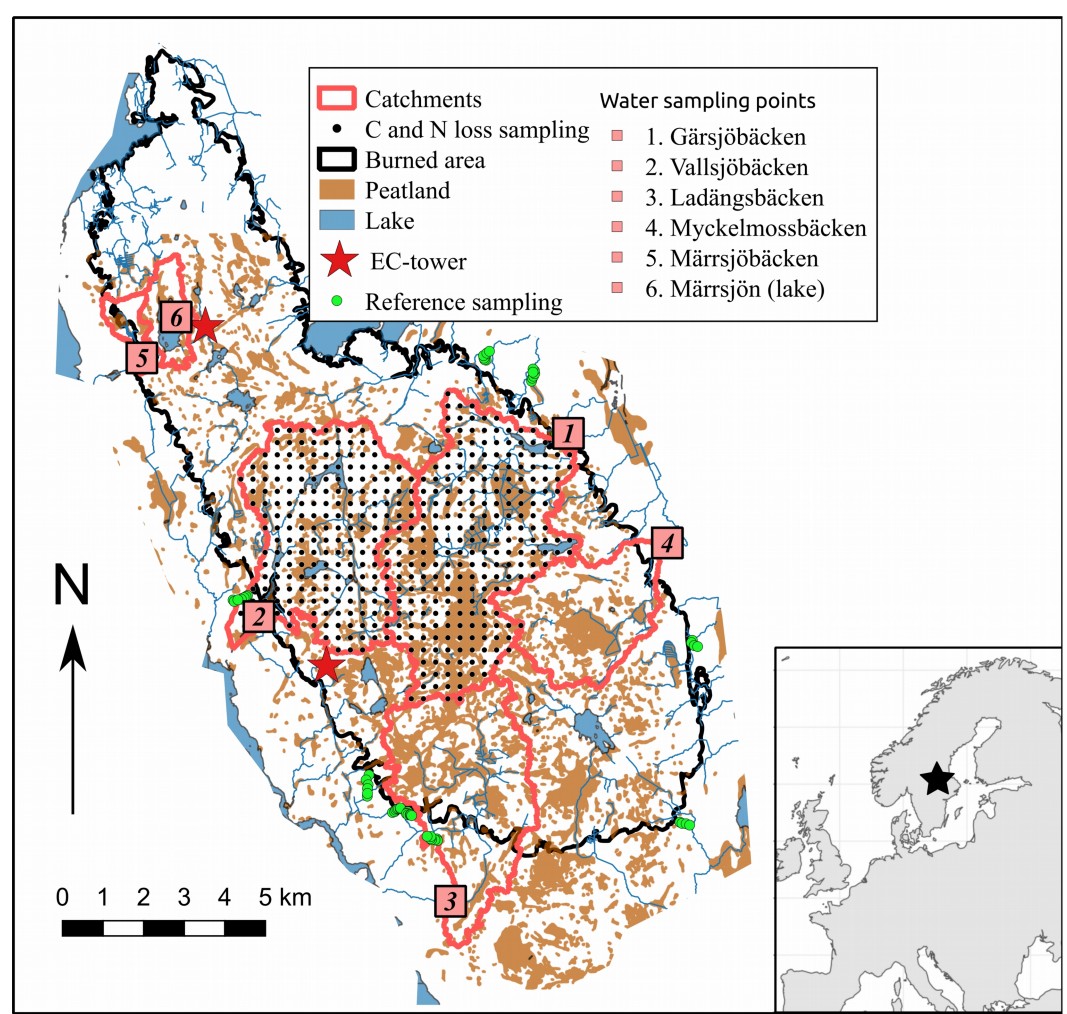

Figure 1: Map over the burnt area showing the sampled catchments, sampling points (terrestrial C and N loss and reference plots, and water sampling stations) and placement of the eddy-covariance towers. Most of the burnt, non-peatland, areas of the catchments Ladängsbäcken, Myckelmossbäcken, and Märrsjöbäcken/Märrsjön, were salvage logged. Table 1 contains information about the catchments.

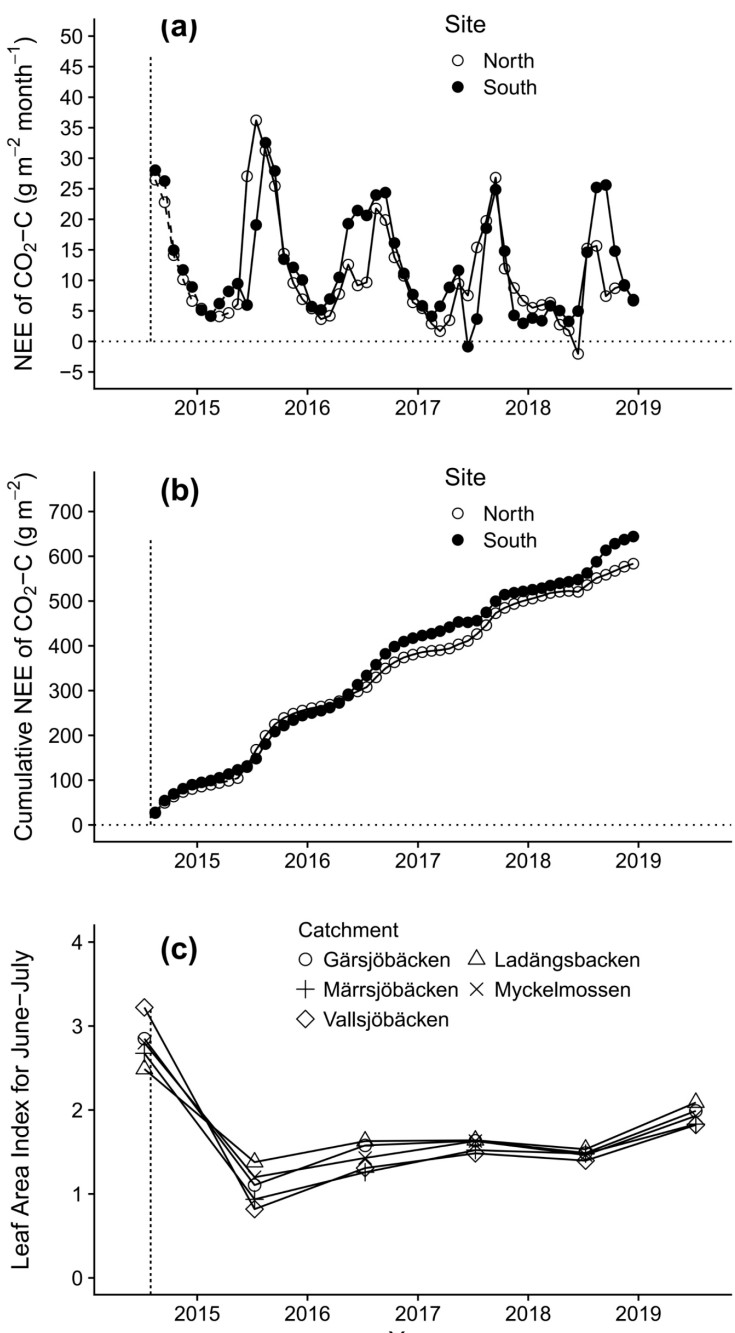

**Figure 2: Carbon fluxes from two burned areas measured by eddy-covariance and changes in Leaf Area Index (LAI) for the five catchments. (a) Monthly CO₂-C fluxes and (b) cumulative CO₂-C flux. Dashed line indicates the period of modelled fluxes. Positive values mean emissions to the atmosphere. (c) Changes in mean summer (June 15 - July 28) LAI over time of the burnt parts of the catchments. First year (2014) shows LAI prior to the fire. Data are from MODIS (500m grid). The fire occurred in august 2014 and is indicated by a dashed vertical.**

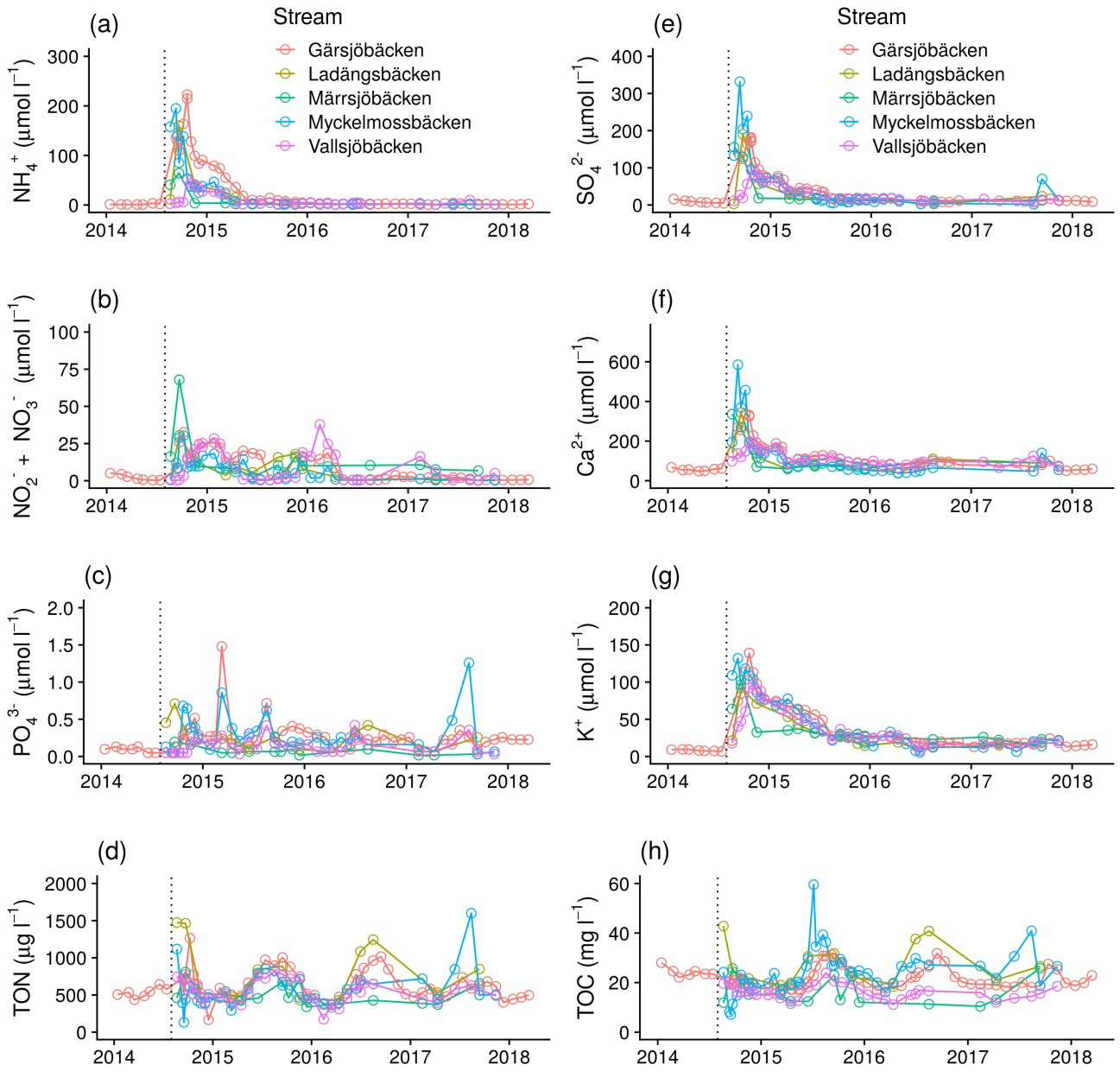

**Figure 3: Temporal changes in concentration of nutrients and major elements in five catchments: (a) NH$_4^+$, (b) NO$_3^-$ + NO$_3^-$, (c) PO$_4^{3-}$, (d) Total Organic N, (e) SO$_4^{2-}$, (f) Ca$^{2+}$, (g) K$^+$, (h) Total Organic C (TOC). Dashed vertical lines indicate the time of the fire.**

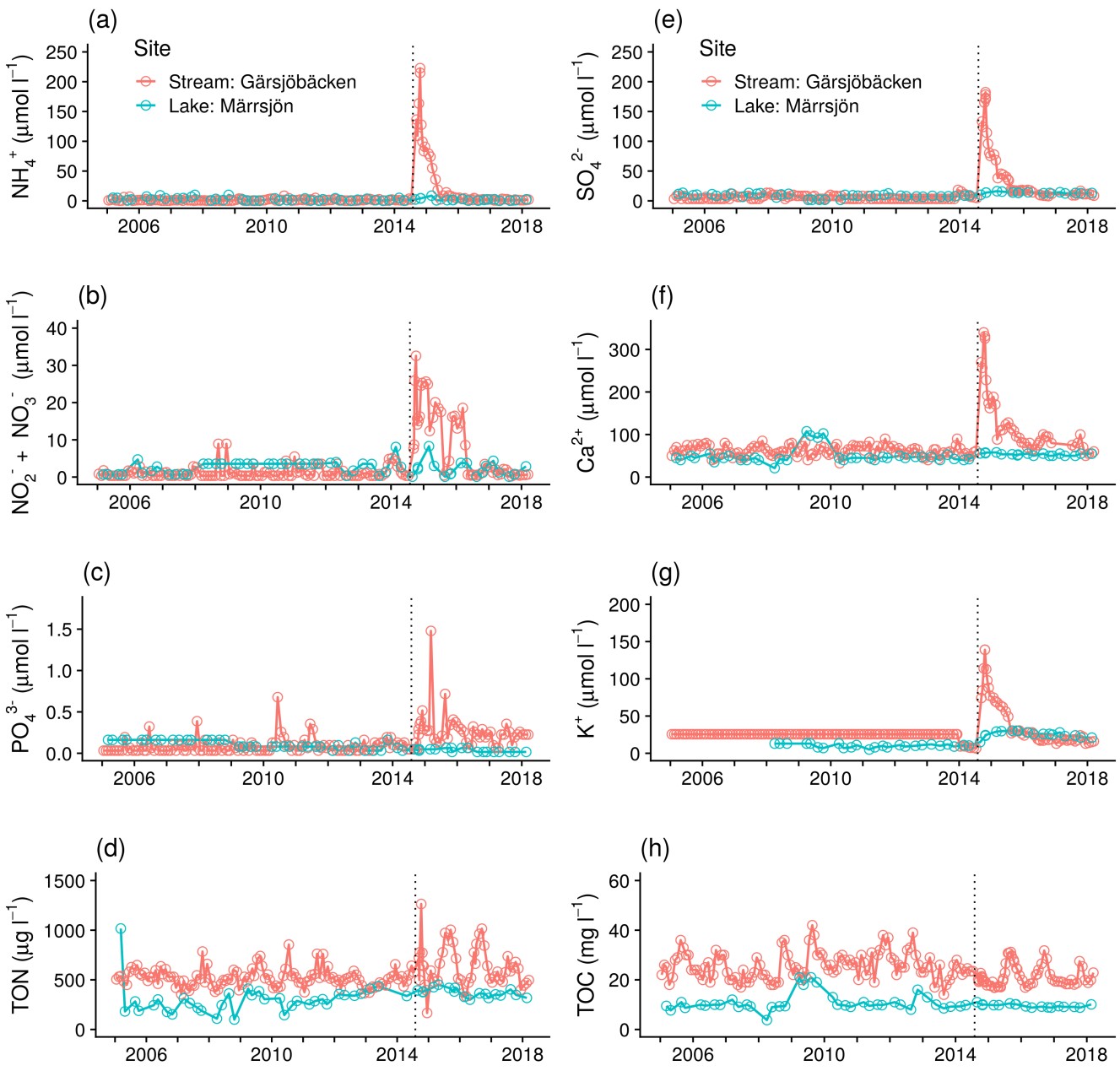

**Figure 4: Long-term (13 years) changes in concentration of nutrients and major elements in Gärsjöbäcken and Märrsjön catchments: (a) NH$_4^+$, (b) NO$_2^-$ + NO$_3^-$, (c) PO$_4^{-3}$, (d) Total Organic N, (e) SO$_4^{-2}$, (f) Ca$^{2+}$, (g) K$^+$,Total Organic C (TOC). Dashed vertical lines indicate the time of the fire. Note that the limit of detection for K$^+$ at Gärsjöbäcken was 25 µmol l$^{-1}$ prior to 2014.**

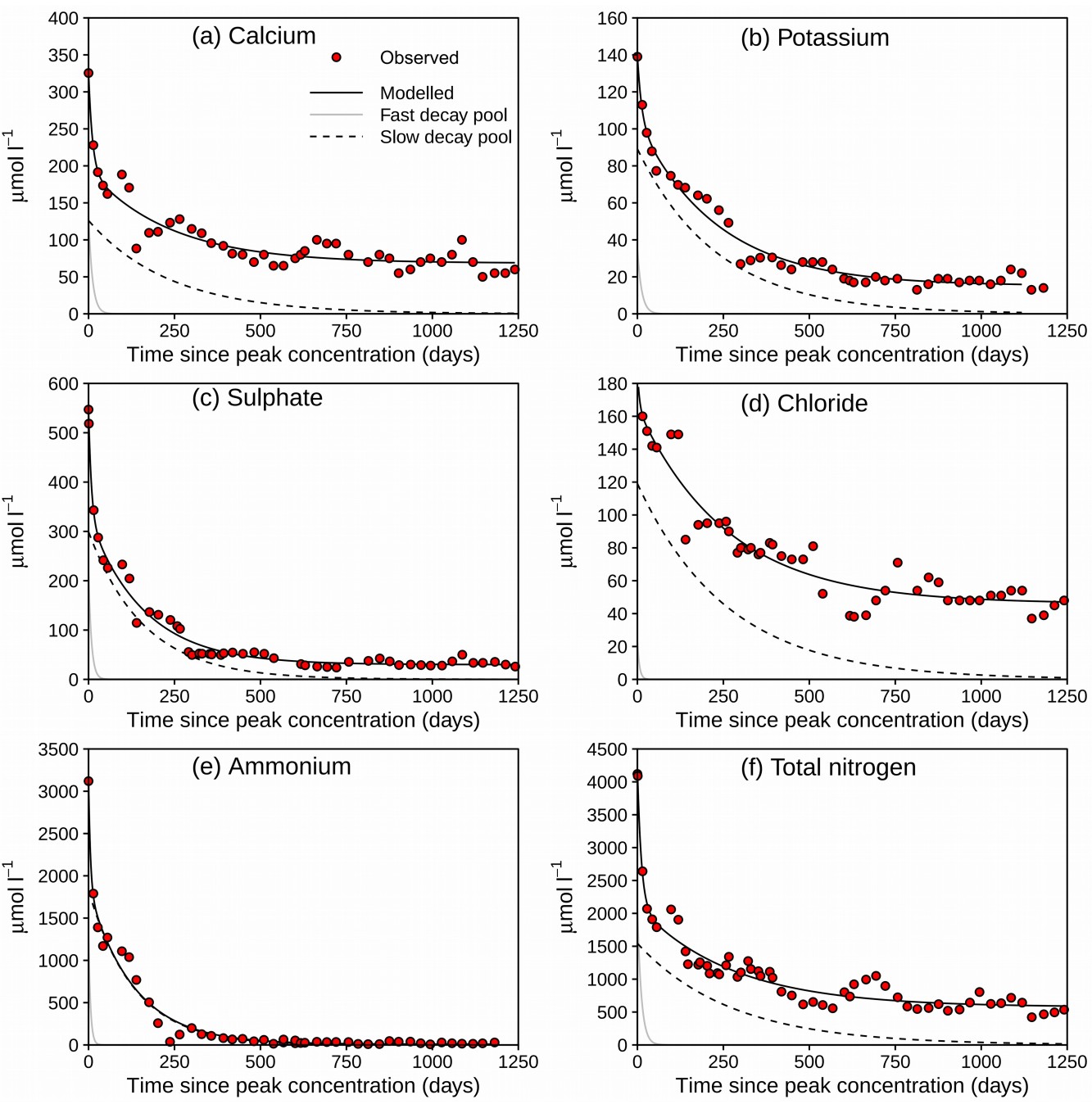

**Figure 5: Fitted decay curves for solutes exhibiting a single concentration peak after the fire, Gärsjöbäcken. See Table 3 for statistics and model fits for the other catchments.**

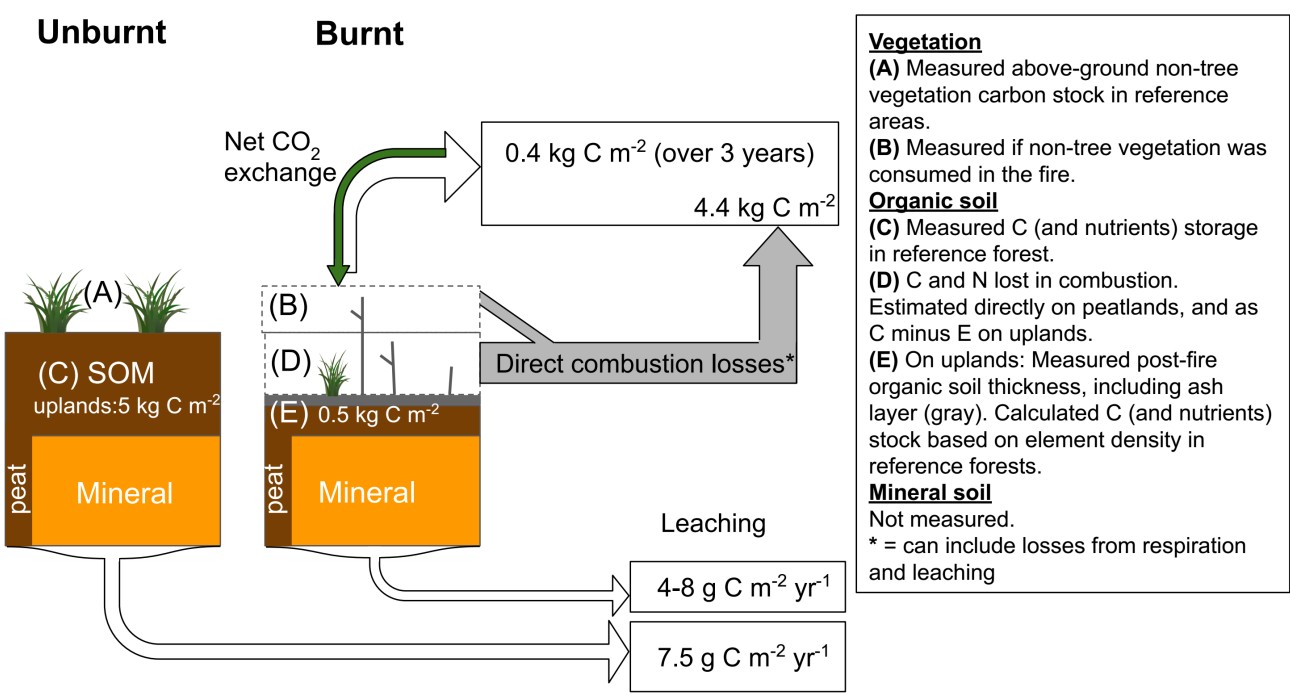

5   **Figure 6: Conceptual C flow diagram during the first 3 years after the fire in a boreal forest catchment. Post-fire leaching is given as a range for the first three years, and leaching from an unburnt catchment (Gärsjöbäcken, Table 2) is the mean of the four years before the fire occurred.**