# Peer review of "The impact of wildfire on biogeochemical fluxes and water quality on boreal catchments"

_Biogeosciences, 2020_

## Referee Comment (RC1) · Anonymous Referee #1 · 30 Nov 2020

The impact of wildfire on biogeochemical fluxes and water quality on boreal catchments

Granath et al., 2020, Biogeosciences Discussions

This study reports on the impacts on wildfire on water quality and $CO_2$ fluxes from a boreal forest catchment in Southern Sweden (which had been monitored pre-fire) by using paired before-after measurements for the decade prior to the fire and three years post-fire to construct elemental budgets. I did enjoy reading this paper and this work appears poised to make a valuable contribution to the literature of the effects of wildfire by leveraging existing pre-fire measurements. As the authors point out, studies on the effects of forest wildfire recovery often lack pre-fire measurements and rely on space-time substitution as a proxy for 'pre-fire' and 'post-fire' conditions, which carries its own set of nebulous assumptions which are avoided in the present study design here. The novel partitioning of post-fire solute fluxes into fast and slow decay pools should be of wide interest as a normalised metric of water quality recovery to baseline post-fire across environments.

It appears the authors have been forthcoming with the history of this manuscript as submitted to a previous journal for peer-review and, as a result, had made substantial revisions and provided a thorough response to previous reviewer comments. I recommend this paper for publication following some primarily minor revisions, focused around language, clarity, and more explicit outline of assumptions and methodological choices throughout.

Abstract: Might be worth including range of study years (including pre-fire monitoring) and year of wildfire in abstract?

Pg 1 Line 18 – 'during the first 12 months' – the first 12 months-post fire?

Pg 1 Line 20 – curious of this terminology, 'ecologically relevant' increases – what criterion is used to determine this? Perhaps (if statistically applicable) 'significant'? Not that statistical testing is required, but if it were carried out, this may be the appropriate venue to specify.

Pg 1 Line 22 – does the partitioning of these pools into 'slow' and 'fast' and the values of these half-lives apply to all analytes?

Pg 1 Line 24 – given this is a study largely of using pattern to infer process, perhaps a stretch to say 'biogeochemical cycles have largely returned to...' and rather best to comment on what precisely was measured in this work, ie, 'dissolved fluxes of nutrients have largely returned to....'

Pg 2 Line 35 – Perhaps best here and throughout introduction/discussion to quantify 'long-term' (one year, ten years, 100 years?) and contextualise in fire return interval for the cited study regions

Pg 3 Line 4 – 'runoff' vs 'run-off' inconsistently stylised throughout

Pg 4 First paragraph – unclear to be how the second half of the first objective (i) "hydrologically exported C, N, S, Ca, K the first three years post-fire," differs from the second objective (i) "post fire water quality trends in five streams...." – are these two separate objectives?

Pg 4 Lines 25 – While topography is certainly a consideration in hydrology this statement might either be reinforced by citation to evidence, or, rather stated as an assumption for watershed delineation, given that in other boreal environments, perhaps 'topography is the last thing to consider' (ie, Devito et al., 2005)

Devito, K., Creed, I., Gan, T., Mendoza, C., Petrone, R., Silins, U., & Smerdon, B. (2005). A framework for broad-scale classification of hydrologic response units on the Boreal Plain: Is topography the last thing to consider? *Hydrological Processes* 19(8), 1705-1714.

Pg 5 Line 6 – Given the attempt in the paper to perform an elemental balance, is there any concern that this first major precipitation event post-fire may have performed some flushing mechanism where a considerable proportion of the post-fire elemental budget for any analyte in this study may have been exported from the catchment while this event was not sampled? Perhaps worthy a caveat in the discussion of why this may or may not be likely?

Pg 5 Line 8 – "high temporal resolution", "longer intervals", "lake was sampled slightly less frequently", here and elsewhere, define each of these precisely. Hourly? Daily? Weekly? Monthly? Was the sampling regularly spaced or focused around precipitation events? Was the sampling design/frequency rooted in literature? Based off or paired with the pre-fire sampling frequency? Given the objective was to estimate export, sampling design can have a significant impact of these estimates (and varies by solute of interest), see for example:

Johnes, P. J. (2007). Uncertainties in annual riverine phosphorus load estimation: Impact of load estimation methodology, sampling frequency, baseflow index and catchment population density. Journal of Hydrology, 332(1-2), 241-258.

Richards, R. P., & Holloway, J. (1987). Monte Carlo studies of sampling strategies for estimating tributary loads. *Water Resources Research*, *23*(10), 1939-1948.

Aulenbach, B. T., Burns, D. A., Shanley, J. B., Yanai, R. D., Bae, K., Wild, A. D., ... & Yi, D. (2016). Approaches to stream solute load estimation for solutes with varying dynamics from five diverse small watersheds. *Ecosphere*, *7*(6), e01298.

Pg 5 Line 21 – What is meant by each 'intersection'? Were the 300 m x 300 m grids divided into subgrids, every, say, 50 or 100 m?

Pg 5 Line 31 – Glad to see the careful considerations and limitations of this method which appears sound and consistent with literature. Is there a quick and transparent back-of-the-envelope calculation that could be included here to contextualise this 'likely small' overestimation of carbon loss (ie, as a potential error) relative to the estimated values, even to just to give a rough order of magnitude, to inform if we are roughly in the territory of, say, 0.1%, 1%, or 10% overestimation?

Pg 8 Line 11 – inconsistent formatting throughout of ions - use of subscripts/superscripts, and including charge, ie NH4 vs $NH_4^+$

Pg 8 Line 18 – What was the basis for model selection following ruling out a single (simple) exponential decay surve? Ie why the partitioning into exactly two pools of fast- and slow-decay superimposed on the baseline – why not three pools and include a 'medium'-decay? Is the two-pool model rooted in literature? Does some information criterion inform that two pools is superior to three (or more) on an added complexity cost analysis? How sensitive would the analysis be to additional complexity?

Pg 9 Line 1 – It appears pH measurements taken to validate this model, but no detail given in methods? Were these measurements in-situ, coincident with the water samples?

Pg 9 Line 8 – Presuming, then, that extended surface water coverage was not an issue at these sites then in terms of pixel removal?

Pg 10 Line 9 – Perhaps for clarity change "Nitrate and ammonium increased…" to "Nitrate and ammonium concentrations in streamflow increased…" and similarly throughout

Pg 10 Line 31 – I am wondering back to the initial question on sampling frequency (Pg 5 Line 8) and how the resolution of sampling overlays with this estimate of the 'fast' decay pool (4-20 days). Would more high-frequency sampling during what seems to have been identified as a critical short-term post-fire period yield finer estimates of this critical period length? Further, is it possible that the omission of the first post-fire precipitation event (Pg 5 Line 6) from the sampling design yielded a considerable portion of this 'fast' pool that was unaccounted?

Pg 11 Lines 2-6 – Were these sequences of inequalities statistically assessed? Perhaps including values of each of these peak/baseline ratios here would be informative and a useful normalised metric for other post-fire studies to compare against.

Page 11 Line 16 – Hanging parentheses

Page 12 Line 5-6 – This may be a stretch to generalise from two studies, if no other annual-basis studies of NEE are available.

Page 13 Line 2 – Is this meant to read 'first year' singular?

Page 14 Line 14 – an interesting observation on similar impacts from such different types of disturbance – what mechanisms would be responsible for these similarities?

Figure 2 – perhaps the fire could be delineated as a horizontal line on the figure as similar to Figure 3?

Figure 6 – this inclusion of methods/assumptions (text on right of figure) is an excellent contribution to laying out the fluxes in an integrated way such as this.

---

## Referee Comment (RC2) · Anonymous Referee #2 · 2 Dec 2020

The impact of wildfire on biogeochemical fluxes and water quality on boreal catchments (Granath et al., 2020, Biogeosciences Discussions) This study reports on the impacts on wildfire on C dynamics and water quality from a boreal forest catchment in Southern Sweden, using paired (before-after) measurements on fire areas. To be honest, I was hoping quite a lot from the paper, as the topic seems really interesting and promising (as both pre fire situation and post-fire conditions were supposed to be included. It would be quite unique possibility to describe quite exactly the C dynamics related with fires (pre fire conditions, combustion, and post-fire conditions), and all this in relatively large scale. Unfortunately, at this stage the paper misses many explanations, and actually entire research is missing some of the needed measurements. Thus, at this stage the authors were not able to convince me that some of their statements are actually

valid. At this stage I have the impression, that by leaving the pyrogenic material measurements (charred material, charcoal, ash) out from the research, the authors are overestimating the C losses through combustion. Also, as the water measurements started weeks after fire (and one week after first rain), the authors are underestimating the fluvial C movements. As the authors have not been explaining how they have been using eddy data (they are presenting net ecosystem exchange (NEE) results, that also includes the photosynthesis (carbon uptake), but they haven't been explaining the proportions of the photosynthesis and respiration, the authors have not been convincing me that their numbers behind different C fluxes are correct. The authors are also completely ignoring the fact (would expect it at least in discussion) that (at least some of) the areas were logged after fires.

Below are my detailed comments: P2 L11: What about Scandinavia? Emissions are bigger or smaller compared to North America, as the fires are completely different in these two regions. P2 L12-13: Compared to what areas? North American areas? Upland soils vs. peatlands? P2 L23-24: New study by Rodríguez-Cardona et al 2020 (Scientific Reports volume 10, Article number: 8722) shows clear post-fire decrease (although they are using longer chronosequences there). P2 L24: What is POC export? P4 L1: It should be stated somewhere here that (at least some) the areas were logged after fire! P4 L1: In intro there is a lot of talk about drained peatlands and/or peatlands. Is this the case also here, are the areas mainly forests on drained peatlands? I think some kind of description of the are would be good to include here. P4 L9: Any expectations/hypothesis? P4 L16-17: Would expect more of the area description. How old was the forest? Was it similar through the area or there was many different stands with different age and tree species? P5 L5-8: Can it be that due to late start you have been actually missing some of the C movement (it is washed through different soil horizons with days after rain)? P5 L28: Is this the same as the "ash layer" mentioned earlier? P5 L30: It can be also up to 60% or even higher (Wiechmann et al 2015. PloS one, 10 (8), e0135014-e0135014). P5 L 30-31: how were the charred logs/snags/stumps treated? If you haven't been measuring the pyrogenic carbon (charcoal, ash) separately, you

are probably overestimating a lot. P6 L9-10: Based on Figure1, these transects and sample plot locations are not similar to the burned area. Please specify how these reference transects were located (how far from each other, etc.). P6 L14: "...three to five soil cores...". Per transect? Per plot? P6 L19-21: If stated like this, then my question is what about Europe and Scandinavia? P7 L4: With eddy, I assume you are measuring net ecosystem exchange (NEE) (including C uptake by photosynthesis and release by respiration). If we assume that everything was killed during fire (but you were saying that at the beginning the fire was not stand replacing) then you would measure the respiration (decomposition, etc.), but the vegetation comes back quite quickly after fire, so I would still say that you are measuring NEE. How you are able to talk about the C emissions? As you are not explaining how you were separate the respiration (C emissions) from the photosynthesis. How big and to what direction is the footprint area of the eddy systems. I would assume that the winds from the west are dominant in these areas, but this way the southern eddy is not measuring fire area (at least most of the time)? Also, the eddys are placed so that you are not able to combine the C loss measurements and eddy data (as they are most probably not overlapping). Any specific reason why the eddys were placed as they were? P7 L18-20: This is really big assumption! Taking also into account that you actually haven't been taking the formed pyrogenic carbon (charcoal, charred material, etc.) into account (not analyzed it separately), your C loss calculations might be overestimated. P9 L13-15: So you are saying that 95% of the C emitted during the fire was coming from O-horizon? You had high severity, stand replacing fires on areas (high intensity), all the trees killed, vegetation removed, and then more than 95% comes from O-horizon? On table 2 there is only one value for emissions during the fire, and no separation by vegetation and/or soil. P9 L23-25: I still think that you were actually missing the biggest fluvial losses (the pyrogenic material that is washed away with first rain event). P10 L1-2: Base on the figures you have been measuring NEE with eddy. Unfortunately, there is no data available about vegetation recovery (biomass, coverage), but I have the impression that 3 years after fire, there is already some new vegetation also in areas with high severity. So one

can't talk anymore about C loss when interpreting the NEE values. P10 L4-5: Now the talk is about C uptake (my previous comment). But the vegetation regrowth data is not presented, and it is still not explained how you separated the respiration and uptake data from each other. P11 L11-12: Sorry, but based on your results and talk, I'm not convinced! By not taking into account (analyzing separately) the amount of charred material and charcoal, you are overestimating the direct emissions from the fires. P11 L16-18: Sorry, but you missed the first rain event (if the first samples were taken week after first post-fire rain), and with that probably also DOC that was washed away (or washed to deeper soil horizons) from the areas. So, I assume you are underestimating the fluvial C loss. P14 L23: Discussion is completely missing the fact that the areas (at least some of them) were logged after fire. How the mixing of soil (pyrogenic material and soil) by machinery would affect the emissions and water quality? How the logging (removing the material that would start to decompose on areas) could affect the C fluxes?

---

## Author Comment (AC2) · 5 Jan 2021

*The impact of wildfire on biogeochemical fluxes and water quality on boreal catchments(Granath et al., 2020, Biogeosciences Discussions) This study reports on the impacts on wildfire on C dynamics and water quality from a boreal forest catchment in Southern Sweden, using paired (before-after) measurements on fire areas. To be honest, I was hoping quite a lot from the paper, as the topic seems really interesting and promising(as both pre fire situation and post-fire conditions were supposed to be included. It would be quite unique possibility to describe quite exactly the C dynamics related withfires (pre fire conditions, combustion, and post-fire conditions), and all this in relatively large scale. Unfortunately, at this stage the paper misses many explanations, and actually entire research is missing some of the needed measure-*

*ments. Thus, at this stage the authors were not able to convince me that some of their statements are actually valid. At this stage I have the impression, that by leaving the pyrogenic material measurements (charred material, charcoal, ash) out from the research, the authors are overestimating the C losses through combustion. Also, as the water measurements started weeks after fire (and one week after first rain), the authors are underestimating the fluvial C movements. As the authors have not been explaining how they have been using eddy data (they are presenting net ecosystem exchange (NEE) results, that also includes the photosynthesis (carbon uptake), but they haven't been explaining the proportions of the photosynthesis and respiration, the authors have not been convincing me that their numbers behind different C fluxes are correct. The authors are also completely ignoring the fact (would expect it at least in discussion) that (at least some of) the areas were logged after fires.*

**RESPONSE:** Thank you for taking the time and performing a very detailed review. We will here respond to the four main listed concerns. These points are partly repeated in the detailed comments but we will sometimes refer to the response here.

Charcoal and other carbon pools: Also reviewer 1 asked about the impact but were less worried that it compromised our conclusions. We do take this issue seriously and have conducted sensitivity analyses to better evaluate the effect of our approach. Fortunately, we have access to data carbon content of the soil charcoal layer. Using data from a recently published study from the same burn (https://doi.org/10.1111/1365-2745.13529, some of the authors were involved in that study as well) we can conclude that the carbon content is roughly 20-25% in this layer. This is lower than the non-burned organic soil, and the bulk density of this charcoal layer suggests that this layer is less compact as well. Consequently, using non-burned organic soil values for carbon bulk density we likely underestimate carbon loss rather than overestimated as we previously stated in the manuscript. If we assume a charcoal layer of 1 cm (reported in older pine forest for studied burn [https://doi.org/10.1111/1365-2745.13529] but for the whole area the thickness is smaller), this underestimation is roughly between 2-45

g C per m2 (or about 0.01-1% of the average calculated loss).

We did not include losses from downed wood as this is a small component in a managed landscape like the one that studied here. The burnt area has around four m3 per hectare of downed wood (Jonsson et al 2016, http://dx.doi.org/10.1016/j.foreco.2016.06.017). With a stem density of 418 kg m-3 for Scots pine (Macdonald, Gardiner and Mason, 2009), and 50% carbon content, the maximum loss from downed wood is about 80 g carbon per m2 (or circa 1.5% of total C loss). However, this maximum value is very unrealistic as downed wood rarely was completely consumed by the fire.

Losses from standing trees were not estimated. It is very hard to make reliable quantifications of such losses (amount of fine branches and needles consumed) and they contribute little to the overall losses in the studied area. For example, after the fire (charred) needles were still present in the burned crowns. We do have data on crown fire severity (crude % scale) that can be used to calculate potential losses from needles and fine branches.

Taken together, in a revised version we aim to better discuss uncertainties and provide potential losses associated with the above carbon pools. If anything, our carbon loss estimates are conservative and not an overestimation.

Early hydrological losses: We agree that there are uncertainties associated with the initial post-fire period, and more sampling points would always be better, but we did not have the time, access/permits or budget to start sampling sooner, or at higher frequency, with no advance warning of the fire. The fact that some solute peaks occurred after our first sampling visit (in some cases two months later) strongly suggests that we did not miss a major flushing event during the immediate post-fire period. We have undertaken a sensitivity analysis of the maximum solute export

that could have occured if an earlier peak had occurred, and the implications of this analysis will be discussed in the manuscript. Here we describe an example of a sensitivity analysis for the Gärsjöbäcken catchment. If we assume that the carbon and nutrient concentration one week after the fire were double the values measured at the first time point (about 3 weeks after), then the impact on the annual fluvial loss is an underestimation of 0.5% for carbon and 1% for nitrogen. This should be viewed as an extreme scenario in our opinion but gives an idea of how small the impact is.

NEE: We focus on NEE as this is the carbon balance and the response of main interest here. We write C emission because NEE showed a net C release. Maybe we should be clearer here and not mix terms, but it is clearly important to note that the ecosystem was losing carbon overall *despite* vegetation regrowth during the first three years post-fire; this suggests a large and sustained loss of carbon from soils and dead organic matter. Either the reviewer has misunderstood how NEE data are interpreted here, or we have misunderstood their point. To try to avoid confusion we will revise the terminology and define NEE in the methods so that it is as clear as possible what these results represent.

Logged areas: We will include a discussion of the potential impact of salvage logging. We didn't see a clear effect and therefore it was not discussed. Note that our two focus catchments were not salvage logged.

*Below are my[reviewer 2] detailed comments:*
*P2 L11: What about Scandinavia? Emissions are bigger or smaller compared to North America, as the fires are completely different in these two regions.*
**RESPONSE:** We are not aware of any data of carbon loss from Scandinavia. Our study is likely one of the first, but we are happy to be corrected.

*P2 L12-13: Compared to what areas? North American areas?Upland soils vs. peatlands?*
**RESPONSE:** Compared to boreal upland soils. This should be clarified.

*P2 L23-24: New study by Rodríguez-Cardona et al 2020(Scientific Reports volume 10, Article number: 8722) shows clear post-fire decrease(although they are using longer chronosequences there).*
**RESPONSE:** Thank you for pointing us to this article. This study has a much longer time-scale (decades), but we see merit in referring to it in the manuscript.

*P2 L24: What is POC export?*
**RESPONSE:** POC=particulate organic carbon. We missed writing out the abbreviation.

*P4 L1: It should be stated somewhere here that (at least some) the areas were logged after fire!*
**RESPONSE:** We are not sure that comparing logged and unlogged areas can be tested in our study, and it was never our intention to do so. That is why we did not add it as a separate question/aim. However, we will discuss this, and consider if it could have influenced post-fire solute behaviour.

*P4 L1: In intro there is a lot of talk about drained peatlands and/or peatlands.Is this the case also here, are the areas mainly forests on drained peatlands? I think some kind of description of the area would be good to include here.*
**RESPONSE:** We presented data on the percentage of open and forested peatlands in Table 1; total peat cover was around 15-30% and yes, most forested peatlands were drained. The catchment attributes were discussed at the start of the methods, but we notice that this part is lacking a reference to Table 1, and we will add that.

*P4 L9: Any expectations/hypothesis?*
**RESPONSE:** Good point. We did have an idea behind testing this and a sentence should be added to better reflect this.

*P4 L16-17: Would expect more of the area description. How old was the forest? Was it similar through the area or there was many different stands with different age and tree species?*
**RESPONSE:** Study area description was a bit too brief. We will certainly expand on

this. In short, forest consisted mostly of even-aged pine dominated stands, varying from clear-cuts up to >100 years forest stands.

*P5 L5-8: Can it be that due to late start you have been actually missing some of the C movement (it is washed through different soil horizons with days after rain)?*
**RESPONSE:** See main response at the beginning.

*P5 L28: Is this the same as the "ash layer" mentioned earlier?*
**RESPONSE:** Yes. We will change this to "charcoal layer" throughout the manuscript.

*P5L30: It can be also up to 60% or even higher (Wiechmann et al 2015. PloS one, 10 (8),e0135014-e0135014). P5 L 30-31: how were the charred logs/snags/stumps treated?If you haven't been measuring the pyrogenic carbon (charcoal, ash) separately, you are probably overestimating a lot.*
**RESPONSE:** Thank you for pointing us to Wiechmann et al 2015. However, they report % C of the charcoal particles and not of the charred/charcoal layer of the organic horizon (O horizon). It is the O horizon that is of interest for us. As we have written in the above response, we now have such data and they show a carbon content around 20-25%. Downed wood was not included in our estimates as this component is rather small in these managed forests (see response above regarding downed wood).

*P6 L9-10: Based on Figure1, these transects and sample plot locations are not similar to the burned area. Please specify how these reference transects were located (how far from each other, etc.).*
**RESPONSE:** The transects were chosen to reflect the variation within the burnt area and we believe we succeeded rather well in placing these transects to achieve this goal. Using forest composition and wetness/topography maps we selected similar combinations of forest types, wetness and topography as found across the burnt area. Originally we wanted to model organic soil depth across the landscape but because the mean organic soil layer varied so little between sampling plots and did not correlate with predictors like soil moisture, we decided to use the mean value.

*P6 L14: "...three to five soil cores...". Per transect? Per plot?*
**RESPONSE:** That should be per plot. Thanks for noticing this.

*P6 L19-21: If stated like this, then my question is what about Europe and Scandinavia?*
**RESPONSE:** We are not aware of any studies estimating organic soil loss during fire in northern Europe (but we might have missed studies of course). The method should not be continent-specific and this can be reworded to describe the method in general, rather than pinpoint it to a specific geographical location.

*P7 L4: With eddy, I assume you are measuring net ecosystem exchange (NEE) (including C uptake by photosynthesis and release by respiration). If we assume that everything was killed during fire (but you were saying that at the beginning the fire was not stand replacing) then you would measure the respiration (decomposition, etc.), but the vegetation comes back quite quickly after fire,so I would still say that you are measuring NEE. How you are able to talk about the C emissions? As you are not explaining how you were separate the respiration (C emissions) from the photosynthesis. How big and to what direction is the footprint area of the eddy systems. I would assume that the winds from the west are dominant in these areas, but this way the southern eddy is not measuring fire area (at least most of the time)? Also, the eddys are placed so that you are not able to combine the Closs measurements and eddy data (as they are most probably not overlapping). Any specific reason why the eddys were placed as they were?*
**RESPONSE:** First, we write (P4, L18) that the fire WAS a stand-replacing fire - i.e. everything died more or less. Second, we do indeed focus on NEE as this is the carbon balance and the response of main interest here. We say C emission because NEE showed a net C release. Maybe we should be clearer here and not mix terms, but it is clearly important to note that the ecosystem was losing carbon overall *despite* vegetation regrowth during the first three years post-fire; this suggests a large and sustained loss of carbon from soils and dead organic matter. Third, towers were 2.5 m high. The southern tower is located about five hundred meters from the unburned forest

and is indeed measuring only over the burnt area. Fourth, the location of the towers were chosen based on proximity to roads (but still in a closed off area to avoid theft of equipment), representativeness of the area, tree height (tall trees near the tower can fall over and damage the equipment). We did not know the exact delineation of the catchments when the towers were set up. The fact that the tower happened to be placed just outside the catchment where we did the carbon loss measurements does not invalidate its use for comparing data; we combined the flux tower measurements, the multiple catchment measurements and the distributed soil measurements to seek to understand whole-ecosystem responses to the fire at the landscape scale. To our knowledge, few if any studies have previously obtained such comprehensive data.

*P7 L18-20: This is really big assumption! Taking also into account that you actually haven't been taking the formed pyrogenic carbon (charcoal, charred material, etc.) into account (not analyzed it separately), your C loss calculations might be overestimated.*
**RESPONSE:** Regarding pyrogenic carbon, see response above. The assumption seems supported in our view. Erosion is negligible in this system and downwards transportation of carbon particles is likely tiny compared to the amount lost in the fire. Other data on soil carbon that we have collected at some selected sites in the same burnt area did not indicate an increase of carbon in the mineral soil, further strengthening our assumption that changes in carbon stock can be ascribed to gaseous emissions (Pérez-Izquierdo et al. 2020 J of Ecology, https://doi.org/10.1111/1365-2745.13529).

*P9 L13-15: So you are saying that 95% of the C emitted during the fire was coming from O-horizon? You had high severity, stand replacing fires on areas (high intensity), all the trees killed, vegetation removed, and then more than 95% comes from O-horizon? On table 2 there is only one value for emissions during the fire, and no separation by vegetation and/or soil.*
**RESPONSE:** In boreal forests the organic soil is a large carbon pool, and most of it was combusted during the fire. As mentioned above, we did not include downed wood in our estimates, but this pool is very small in managed forests. We also did not estimate loss

from trees (needles, fine branches). The 95% statement was comparing belowground and the forest floor. This will be corrected and we will discuss the potential contribution from downed wood and trees (see earlier response). Table 2 only gives the sum as we wanted to focus on the overall picture. However as the review has queried this interpretation we can provide disaggregated estimates of carbon loss in the text.

*P9 L23-25: I still think that you were actually missing the biggest fluvial losses (the pyrogenic material that is washed away with first rain event).*
**RESPONSE:** See our response earlier.

*P10 L1-2: Base on the figures you have been measuring NEE with eddy. Unfortunately, there is no data available about vegetation recovery (biomass, coverage), but I have the impression that 3 years after fire, there is already some new vegetation also in areas with high severity. So one can't talk anymore about C loss when interpreting the NEE values.*
**RESPONSE:** Yes, for sure there was vegetation recovery in the flux tower footprints, and throughout the study area. The flux towers measured this as part of the NEE, i.e. the measurements represent the balance of vegetation carbon gain and ongoing soil and biomass carbon loss. These measurements clearly show sustained positive NEE over the 3 years post-fire, i.e. the ecosystem was a net source of $CO_2$ to the atmosphere despite vegetation regrowth. Either the reviewer has misunderstood how NEE data are interpreted here, or we have misunderstood their point. To try to avoid confusion we will revise the terminology here and define NEE in the methods so that it is as clear as possible what these results represent.

*P10 L4-5: Now the talk is about C uptake (my previous comment). But the vegetation regrowth data is not presented, and it is still not explained how you separated the respiration and uptake data from each other.*
**RESPONSE:** Vegetation growth is presented as leaf area increase. We are not sure we follow the reviewers comment here as we do say "net carbon uptake", i.e. the balance of vegetation growth and soil/dead biomass loss as noted above. We did not attempt

to separate uptake and respiration in our presentation of the data.

*P11 L11-12: Sorry, but based on your results and talk, I'm not convinced! By not taking into account (analyzing separately) the amount of charred material and charcoal, you are overestimating the direct emissions from the fires.*
**RESPONSE:** See earlier response.

*P11 L16-18: Sorry, but you missed the first rain event (if the first samples were taken week after first post-fire rain), and with that probably also DOC that was washed away (or washed to deeper soil horizons) from the areas. So, I assume you are underestimating the fluvial C loss.*
**RESPONSE:** See earlier response on the early post-fire period.

*P14 L23: Discussion is completely missing the fact that the areas(at least some of them) were logged after fire. How the mixing of soil (pyrogenic material and soil) by machinery would affect the emissions and water quality? How the logging (removing the material that would start to decompose on areas) could affect the C fluxes?'*
**RESPONSE:** It is correct that a large portion of two catchments were logged, and one other catchment experienced some logging. Only older stands were salvage logged. Note that the two focus catchments were not logged (except a tiny part on one edge of the catchment and along some roads). Logging started in general in the spring the year after the fire (in the manuscript it says logging was done within 3-6 months after the fire, but this is incorrect for the investigated catchments). Interestingly, when examining the water chemistry the logged catchment does not stand out much from the other catchments. The impact may actually be smaller than expected (the absence of extreme topography may have limited the impact). Removal of singed older trees probably had little impact on carbon emission at the site over the first years. While the trees were killed, most of the stemwood remained intact after the fire (and most trees that were rooted into mineral soil remained standing). This woody material is slow to decompose (particularly when singed), and (in areas that were not salvage logged) it was still present by the end of our study period. The gradual decay of this material

and charred needles will have contributed to measured NEE in these areas, which our results suggest made a relatively modest contribution to overall carbon loss during the three years of measurements. Clearly, decomposition of dead biomass will continue to contribute to $CO_2$ loss for many years to come, so the proportional contribution of this C pool to total losses might be expected to gradually increase over time; we have noted this in the revised manuscript. Where salvage logging occured it is clearly a more rapid and substantial pathway for C loss as the wood is removed. But again, our two focus catchments did not experience logging.

---

## Author Comment (AC3) · 5 Jan 2021

*This study reports on the impacts on wildfire on water quality and CO 2 fluxes from a boreal forest catchment in Southern Sweden (which had been monitored pre-fire) by using paired before-after measurements for the decade prior to the fire and three years post-fire to construct elemental budgets. I did enjoy reading this paper and this work appears poised to make a valuable contribution to the literature of the effects of wildfire by leveraging existing pre-fire measurements. As the authors point out, studies on the effects of forest wildfire recovery often lack pre-fire measurements and rely on space-time substitution as a proxy for 'pre-fire' and 'post-fire' conditions, which carries its own set of nebulous assumptions which are avoided in the present study design here. The novel partitioning of post-fire solute fluxes into fast and slow decay*

[Figure]

*pools should be of wide interest as a normalised metric of water quality recovery to baseline post-fire across environments. It appears the authors have been forthcoming with the history of this manuscript as submitted to a previous journal for peer-review and, as a result, had made substantial revisions and provided a thorough response to previous reviewer comments. I recommend this paper for publication following some primarily minor revisions, focused around language, clarity, and more explicit outline of assumptions and methodological choices throughout.*

**RESPONSE:** Thank you for reviewing our manuscript and we are happy to see that you found our study valuable. In our revision we will aim to improve language and expand the text around the methods used (also needed to respond to criticism raised by reviewer 2).

*Abstract: Might be worth including range of study years (including pre-fire monitoring) and year of wildfire in abstract?*
**RESPONSE:** We agree.

*Pg 1 Line 18 – 'during the first 12 months' – the first 12 months-post fire?*
**RESPONSE:** Correct. Will be added.

*Pg 1 Line 20 – curious of this terminology, 'ecologically relevant' increases – what criterion is used to determine this? Perhaps (if statistically applicable) 'significant'? Not that statistical testing is required, but if it were carried out, this may be the appropriate venue to specify.*
**RESPONSE:** What we mean is a change that potentially can have a non-negligible impact on biota. This is based on joint judgement of previous studies but no specific criteria are used. The point is that the change is not tiny and we chose this terminology instead of listing all the numbers of these elements.

*Pg 1 Line 22 – does the partitioning of these pools into 'slow' and 'fast' and the values of these half-lives apply to all analytes?*
**RESPONSE:** Yes, at least for the ones we tested. We will clarify which analytes we

tested.

*Pg 1 Line 24 – given this is a study largely of using pattern to infer process, perhaps a stretch to say 'biogeochemical cycles have largely returned to...' and rather best to comment on what precisely was measured in this work, ie, 'dissolved fluxes of nutrients have largely returned to....'*
**RESPONSE:** "Biogechemical cycles" is probably a stretch. We will reword this and the suggestion is good.

*Pg 2 Line 35 – Perhaps best here and throughout introduction/discussion to quantify 'long-term' (one year, ten years, 100 years?) and contextualise in fire return interval for the cited study regions*
**RESPONSE:** Good point. Long/short-term is ambiguous and should be better defined. We will check the manuscript and improve clarity on this. Fire return intervals may be harder to provide (varies over time and space) but will try to do so when possible and relevant.

*Pg 3 Line 4 – 'runoff' vs 'run-off' inconsistently stylised throughout*
**RESPONSE:** Will be fixed.

*Pg 4 First paragraph – unclear to be how the second half of the first objective (i) "hydrologically exported C, N, S, Ca, K the first three years post-fire," differs from the second objective (i) "post fire water quality trends in five streams...." – are these two separate objectives?*
**RESPONSE:** The first part here, "hydrologically exported", refers to the total amount while the second part focuses on concentrations. It is probably clearer if we add "the amount of hydrologically...".

*Pg 4 Lines 25 – While topography is certainly a consideration in hydrology this statement might either be reinforced by citation to evidence, or, rather stated as an assumption for watershed delineation, given that in other boreal environments, perhaps 'topography is the last thing to consider' (ie, Devito et al., 2005)Devito, K., Creed, I.,*

[Figure]

*Gan, T., Mendoza, C., Petrone, R., Silins, U., Smerdon, B. (2005). A framework for broad-scale classification of hydrologic response units on the Boreal Plain: Is topography the last thing to consider? Hydrological Processes 19(8), 1705-1714.*

**RESPONSE:** Using topography for watershed delineation is not exact but it should work pretty well. The paper by Devito et al 2005 is focusing on the whole hydrological response (and the controls) and for flow and balance we are employing a well-tested model (S-hype) which indeed considers other factors. In a revised version this should be more developed than in the current version.

*Pg 5 Line 6 – Given the attempt in the paper to perform an elemental balance, is there any concern that this first major precipitation event post-fire may have performed some flushing mechanism where a considerable proportion of the post-fire elemental budget for any analyte in this study may have been exported from the catchment while this event was not sampled? Perhaps worthy a caveat in the discussion of why this may or may not be likely?*

**RESPONSE:** This was also raised by reviewer 2. We agree that ideally you want to start to sample the day the fire has been put out. However, logistically this is rarely possible (if we have missed studies that have done this we would like to know to get an idea how large concentrations can be). There are a few reasons why these first 2-3 weeks are unlikely to be important. First, the amount of precipitation was not very large. Second, some catchments showed their concentration peak a few weeks after the first sampling point, indicating that flushing (at a catchment scale) often is delayed due to buffering in the system. To further strengthen our assumption that this first period had a small impact on our results, we have performed a sensitivity analysis and re-calculated an upper estimate of the amounts that could have been exported if the flush started earlier. The implications of this analysis will be discussed in the manuscript. Here we describe an example of the sensitivity analysis for the Gärsjöbäcken catchment. If we assume that the carbon and nutrient concentrations one week after the fire were double the values measured as the first time point (about 3 weeks after), then the impact on the annual budget is an underestimation of 0.5% for carbon and 1% for nitrogen. This

should be viewed as an extreme (unrealistic) scenario in our opinion but gives an idea of how small the impact is.

*Pg 5 Line 8 – "high temporal resolution", "longer intervals", "lake was sampled slightly less frequently", here and elsewhere, define each of these precisely. Hourly? Daily? Weekly? Monthly? Was the sampling regularly spaced or focused around precipitation events? Was the sampling design/frequency rooted in literature? Based off or paired with the pre-fire sampling frequency? Given the objective was to estimate export, sampling design can have a significant impact of these estimates (and varies by solute of interest), see for example: Johnes, P. J. (2007). Uncertainties in annual riverine phosphorus load estimation: Impact of load estimation methodology, sampling frequency, baseflow index and catchment population density. Journal of Hydrology, 332(1-2), 241-258. Richards, R. P., Holloway, J. (1987). Monte Carlo studies of sampling strategies for estimating tributary loads. Water Resources Research, 23(10), 1939-1948. Aulenbach, B. T., Burns, D. A., Shanley, J. B., Yanai, R. D., Bae, K., Wild, A. D., ... Yi, D. (2016). Approaches to stream solute load estimation for solutes with varying dynamics from five diverse small watersheds. Ecosphere, 7(6), e01298.*

**RESPONSE:** Thank you for pointing out the poor description of our approach to estimated loads. Sampling design was aimed to start "as often as possible" after the fire (of course it is not easy to quickly set up the sampling with limited resources and time, and wildfires are intrinsically hard to plan for) and ranged from a few weeks at the start after the fire to more like monthly. Some consideration was taken to capture potential peaks (for example spring flood). Given that it appears that we captured the post-fire decline in concentration fairly well, we think the sampling intensity was sufficient to produce estimates with good precision. We estimate that annual loads should not be off by more than 5-10% (based on Aulenbach et al. 2016). Our overall approach, using a period-weighted method to estimate load, is what is recommended by Aulenbach et al. (2016) when there is a weak concentration - discharge relationship. In a revised version these things will be properly explained and referenced, and we will also add a discussion on how large our load estimates can be.

*Pg 5 Line 21 – What is meant by each 'intersection'? Were the 300 m x 300 m grids divided into subgrids, every, say, 50 or 100 m?*
**RESPONSE:** Poorly worded. We mean each grid point (i.e., 300 m between each sampling point).

*Pg 5 Line 31 – Glad to see the careful considerations and limitations of this method which appears sound and consistent with literature. Is there a quick and transparent back-of-the-envelope calculation that could be included here to contextualise this 'likely small' overestimation of carbon loss (ie, as a potential error) relative to the estimated values, even to just to give a rough order of magnitude, to inform if we are roughly in the territory of, say, 0.1%, 1%, or 10% overestimation?*
**RESPONSE:** Yes, it is our understanding that this is the normal approach as it should have a minor impact compared to other sources of errors but is time-consuming to estimate accurately. Reviewer 2 expressed serious concerns about this and we have indeed tried to run some sensitivity analyses to estimate the effect. We used published data on charcoal carbon content and charcoal weight from another study from the same burnt area (Perez-Izquierdo et al 2020 J of Ecology). We can now show that we likely underestimate carbon loss by treating this thin charcoal layer as an organic soil, but only with maximum 45 g m-2 (or roughly 1% of the total loss).

*Pg 8 Line 11 – inconsistent formatting throughout of ions - use of sub-scripts/superscripts, and including charge, ie NH4 vs NH 4+*
**RESPONSE:** we should probably keep it to the correct $NH_4^+$.

*Pg 8 Line 18 – What was the basis for model selection following ruling out a single (simple) exponential decay surve? Ie why the partitioning into exactly two pools of fast- and slow-decay superimposed on the baseline – why not three pools and include a 'medium'-decay? Is the two-pool model rooted in literature? Does some information criterion inform that two pools is superior to three (or more) on an added complexity cost analysis? How sensitive would the analysis be to additional complexity?*
**RESPONSE:** The two pool model was based on observed solute behaviour; most

showed a period of very rapid decline from the immediate post-fire peak, followed by a more gradual decline to baseline levels over around a year. A single-exponential model was unable to reproduce both the rapid initial decline and the longer-term decrease, whereas a two-pool model generally gave a good fit to multiple solutes (e.g. Figure 5) and appeared to be mechanistically interpretable, as discussed. A three (or more) pool model would have over-fitted the data. To our knowledge, the two-pool approach to post-fire solute behaviour is new (and thus not rooted in the literature) but we believe it offers some valuable mechanistic insights, and may be of value to other researchers in future. We will expand our justification for the approach and the discussion of its wider application in the revised version.

*Pg 9 Line 1 – It appears pH measurements taken to validate this model, but no detail given in methods? Were these measurements in-situ, coincident with the water samples?*
**RESPONSE:** Yes, coincident with the water sample. This will be added.

*Pg 9 Line 8 – Presuming, then, that extended surface water coverage was not an issue at these sites then in terms of pixel removal?*
**RESPONSE:** Sorry but we don't understand this comment.

*Pg 10 Line 9 – Perhaps for clarity change "Nitrate and ammonium increased..." to "Nitrate and ammonium concentrations in streamflow increased..." and similarly throughout*
**RESPONSE:** Good point. We should be more specific what we mean.

*Pg 10 Line 31 – I am wondering back to the initial question on sampling frequency (Pg 5 Line 8) and how the resolution of sampling overlays with this estimate of the 'fast' decay pool (4-20 days). Would more high-frequency sampling during what seems to have been identified as a critical short-term post-fire period yield finer estimates of this critical period length? Further, is it possible that the omission of the first post-fire precipitation event (Pg 5 Line 6) from the sampling design yielded a considerable portion*

[Figure]

*of this 'fast' pool that was unaccounted?*

**RESPONSE:** We agree that there are uncertainties associated with the initial post-fire period, and more sampling points would always be better, but as noted above we did not have the time, access/permits or budget to start sampling sooner, or at higher frequency, with no advance warning of the fire. The fact that some solute peaks occurred after our first sampling visit (in some cases two months later) strongly suggests that we did not miss a major flushing event during the immediate post-fire period. As discussed elsewhere we have undertaken a sensitivity analysis of the maximum solute export that could have occured if an earlier peak had occurred (the maximal potential impact on the annual fluvial loss is probably an underestimation of 0.5% for carbon and 1% for nitrogen), and the implications of this analysis are discussed.

*Pg 11 Lines 2-6 – Were these sequences of inequalities statistically assessed? Perhaps including values of each of these peak/baseline ratios here would be informative and a useful normalised metric for other post-fire studies to compare against.*

**RESPONSE:** Good idea and we did consider it ourselves. However, with only 5 catchments a statistical evaluation seems unwarranted. We can check if there is a metric that can be used to normalise, e.g. mean catchment residence time can be another option.

*Page 11 Line 16 – Hanging parentheses*
**RESPONSE:** Thanks for noticing.

*Page 12 Line 5-6 – This may be a stretch to generalise from two studies, if no other annual-basis studies of NEE are available.*
**RESPONSE:** Will be reformulated to "These values are strikingly similar to our two sites (155 to 165 g C m $-2$ yr $-1$ over two years), but further research is needed to establish if such values are typical for boreal uplands post-fire."

*Page 13 Line 2 – Is this meant to read 'first year' singular?*
**RESPONSE:** Good catch. Should be "first year".

*Page 14 Line 14 – an interesting observation on similar impacts from such different types of disturbance– what mechanisms would be responsible for these similarities?*
**RESPONSE:** The main reason would be less plant uptake and sometimes in combination of increased mineralisation. This can be briefly mentioned in the discussion.

*Figure 2 – perhaps the fire could be delineated as a horizontal line on the figure as similar to Figure 3?*
**RESPONSE:** Yes, we can add that for clarity.

*Figure 6 – this inclusion of methods/assumptions (text on right of figure) is an excellent contribution to laying out the fluxes in an integrated way such as this.*
**RESPONSE:** Thank you. Even if it has limitation, we think our box diagram helps our understanding of the main post-fire nutrient and carbon flow paths.

---

## Author Response (AR1)

**Reviewer 1**

*This study reports on the impacts on wildfire on water quality and CO 2 fluxes from a boreal forest catchment in Southern Sweden (which had been monitored pre-fire) by using paired before-after measurements for the decade prior to the fire and three years post-fire to construct elemental budgets. I did enjoy reading this paper and this work appears poised to make a valuable contribution to the literature of the effects of wildfire by leveraging existing pre-fire measurements. As the authors point out, studies on the effects of forest wildfire recovery often lack pre-fire measurements and rely on space-time substitution as a proxy for pre-fire and post-fire conditions, which carries its own set of nebulous assumptions which are avoided in the present study design here. The novel partitioning of post-fire solute fluxes into fast and slow decay pools should be of wide interest as a normalised metric of water quality recovery to baseline post-fire across environments. It appears the authors have been forthcoming with the history of this manuscript as submitted to a previous journal for peer-review and, as a result, had made substantial revisions and provided a thorough response to previous reviewer comments. I recommend this paper for publication following some primarily minor revisions, focused around language, clarity, and more explicit outline of assumptions and methodological choices throughout.*

**RESPONSE:** Thank you for reviewing our manuscript and we are happy to see that you found our study valuable. In our revision have aimed to improve language and expanded the text around the methods used (also needed to respond to criticism raised by reviewer no 2).

*Abstract: Might be worth including range of study years (including pre-fire monitoring) and year of wildfire in abstract?*
**RESPONSE:** Added.

*Pg 1 Line 18 during the first 12 months the first 12 months-post fire?*
**RESPONSE:** Correct. Added.

*Pg 1 Line 20 curious of this terminology, ecologically relevant increases what criterion is used to determine this? Perhaps (if statistically applicable) significant? Not that statistical testing is required, but if it were carried out, this may be the appropriate venue to specify.*

**RESPONSE:** What we mean is a change that potentially can have a non-negligible impact on biota. This is based on joint judgement of previous studies but no specific criteria are used. The point is that the change is not tiny and we chose this terminology instead of listing all the numbers of these elements.

*Pg 1 Line 22  does the partitioning of these pools into slow and fast and the values of these half-lives apply to all analytes?*
**RESPONSE:** Yes, at least for the ones we tested. We have clarified which analytes we tested.

*Pg 1 Line 24  given this is a study largely of using pattern to infer process, perhaps a stretch to say biogeochemical cycles have largely returned to... and rather best to comment on what precisely was measured in this work, ie, dissolved fluxes of nutrients have largely returned to....*
**RESPONSE:** Biogechemical cycles is probably a stretch. We have reword this according to the suggestion.

*Pg 2 Line 35  Perhaps best here and throughout introduction/discussion to quantify long-term (one year, ten years, 100 years?) and contextualise in fire return interval for the cited study regions*
**RESPONSE:** Good point. Long/short-term is ambiguous and should be better defined. We checked the manuscript and improved clarity on this in a few places. Fire return intervals is harder to provide (varies over time and space).

*Pg 3 Line 4  runoff vs run-off inconsistently stylised throughout*
**RESPONSE:** Fixed.

*Pg 4 First paragraph  unclear to be how the second half of the first objective (i) hydrologically exported C, N, S, Ca, K the first three years post-fire, differs from the second objective (i) post fire water quality trends in five streams.... are these two separate objectives?*
**RESPONSE:** The first part here, hydrologically exported, refers to the total amount while the second part focuses on concentrations. Changed to the amount of hydrologically.

*Pg 4 Lines 25  While topography is certainly a consideration in hydrology*

*this statement might either be reinforced by citation to evidence, or, rather stated as an assumption for watershed delineation, given that in other boreal environments, perhaps topography is the last thing to consider (ie, Devito et al., 2005)Devito, K., Creed, I., Gan, T., Mendoza, C., Petrone, R., Silins, U., Smerdon, B. (2005). A framework for broadscale classification of hydrologic response units on the Boreal Plain: Is topography the last thing to consider? Hydrological Processes 19(8), 1705-1714.*

**RESPONSE:** Using topography for watershed delineation is not exact but it should work pretty well. The paper by Devito et al 2005 is focusing on the whole hydrological response (and the controls) and for flow and balance we are employing a well-tested model (S-hype) which indeed considers other factors as briefly described in our manuscript.

*Pg 5 Line 6 Given the attempt in the paper to perform an elemental balance, is there any concern that this first major precipitation event post-fire may have performed some flushing mechanism where a considerable proportion of the post-fire elemental budget for any analyte in this study may have been exported from the catchment while this event was not sampled? Perhaps worthy a caveat in the discussion of why this may or may not be likely?*

**RESPONSE:** This was also raised by reviewer 2. We agree that ideally you want to start to sample the day the fire has been put out. However, logistically this is rarely possible (if we have missed studies that have done this we would like to know to get an idea how large concentrations can be). There are a few reasons why these first 2-3 weeks are unlikely to be important. First, the amount of precipitation was not very large. Second, some catchments showed their concentration peak a few weeks after the first sampling point, indicating that flushing (at a catchment scale) often is delayed due to buffering in the system. To further strengthen our assumption that this first period had a small impact on our results, we have performed a sensitivity analysis and re-calculated an upper estimate of the amounts that could have been exported if the flush started earlier. We have included a sensitivity analysis for the Gärsjöbäcken catchment in the Results. If we assume that the carbon and nutrient concentrations one week after the fire were double the values measured as the first time point (about 3 weeks after), then the impact on the annual budget is an underestimation of 0.5% for carbon and 1% for nitrogen. This should be viewed as an extreme (unrealistic) scenario in our opinion but gives an idea of how small the impact is.

*Pg 5 Line 8 high temporal resolution, longer intervals, lake was sampled slightly less frequently, here and elsewhere, define each of these precisely. Hourly? Daily? Weekly? Monthly? Was the sampling regularly spaced or focused around precipitation events? Was the sampling design/frequency rooted in literature? Based off or paired with the pre-fire sampling frequency? Given the objective was to estimate export, sampling design can have a significant impact of these estimates (and varies by solute of interest), see for example: Johnes, P. J. (2007). Uncertainties in annual riverine phosphorus load estimation: Impact of load estimation methodology, sampling frequency, baseflow index and catchment population density. Journal of Hydrology, 332(1-2), 241-258. Richards, R. P., Holloway, J. (1987). Monte Carlo studies of sampling strategies for estimating tributary loads. Water Resources Research, 23(10), 1939-1948. Aulenbach, B. T., Burns, D. A., Shanley, J. B., Yanai, R. D., Bae, K., Wild, A. D., ... Yi, D. (2016). Approaches to stream solute load estimation for solutes with varying dynamics from five diverse small watersheds. Ecosphere, 7(6), e01298.*

**RESPONSE:** Thank you for pointing out the poor description of our approach to estimated loads. Sampling design was aimed to start as often as possible after the fire (of course it is not easy to quickly set up the sampling with limited resources and time, and wildfires are intrinsically hard to plan for) and ranged from a few weeks at the start after the fire to more like monthly. Some consideration was taken to capture potential peaks (for example spring flood). Given that it appears that we captured the post-fire decline in concentration fairly well, we think the sampling intensity was sufficient to produce estimates with good precision. We estimate that annual loads should not be off by more than 5-10% (based on Aulenbach et al. 2016). Our overall approach, using a periodweighted method to estimate load, is what is recommended by Aulenbach et al. (2016) when there is a weak concentration - discharge relationship. In a revised version these things are better explained and referenced in the Methods, and we also added the expected error in the table caption of table 2.

*Pg 5 Line 21 What is meant by each intersection? Were the 300 m x 300 m grids divided into subgrids, every, say, 50 or 100 m?*

**RESPONSE:** Poorly worded. We mean each grid point (i.e., 300 m between each sampling point). Changed in the manuscript.

*Pg 5 Line 31 Glad to see the careful considerations and limitations of this method which appears sound and consistent with literature. Is there a quick and transparent back-of-the-envelope calculation that could be included here to contextualise this likely small overestimation of carbon loss (ie, as a potential error) relative to the estimated values, even to just to give a rough order of magnitude, to inform if we are roughly in the territory of, say, 0.1%, 1%, or 10% overestimation?*

**RESPONSE:** Yes, it is our understanding that this is the normal approach as it should have a minor impact compared to other sources of errors but is time-consuming to estimate accurately. Reviewer 2 expressed serious concerns about this and we have now added a sensitivity analyses in the Results to estimate the effect. We used published data on ash carbon content and ash weight from another study from the same burnt area (Perez-Izquierdo et al 2020 J of Ecology). We can now show that we likely underestimate carbon loss by treating this thin charcoal layer as an organic soil, but only with maximum 45 g m-2 (or roughly 1% of the total losses).

*Pg 8 Line 11 inconsistent formatting throughout of ions - use of subscripts/superscripts, and including charge, ie NH4 vs NH 4+*

**RESPONSE:** we should probably keep it to the correct NH4+.

*Pg 8 Line 18 What was the basis for model selection following ruling out a single (simple) exponential decay surve? Ie why the partitioning into exactly two pools of fast- and slow-decay superimposed on the baseline why not three pools and include a medium-decay? Is the two-pool model rooted in literature? Does some information criterion inform that two pools is superior to three (or more) on an added complexity cost analysis? How sensitive would the analysis be to additional complexity?*

**RESPONSE:** The two pool model was based on observed solute behaviour; most showed a period of very rapid decline from the immediate post-fire peak, followed by a more gradual decline to baseline levels over around a year. A single-exponential model was unable to reproduce both the rapid initial decline and the longer-term decrease, whereas a two-pool model generally gave a good fit to multiple solutes (e.g. Figure 5) and appeared to be mechanistically interpretable, as discussed. A three (or more) pool model would have over-fitted the data. To our knowledge, the two-pool approach to post-fire solute behaviour is new (and thus not rooted in the literature but known from organic matter decomposition theory) and we believe it offers

some valuable mechanistic insights, and may be of value to other researchers in future. We have edited the text and a reference in our aims and expanded the discussion a bit.

*Pg 9 Line 1  It appears pH measurements taken to validate this model, but no detail given in methods? Were these measurements in-situ, coincident with the water samples?*
**RESPONSE:** Yes, coincident with the water sample. This has been added.

*Pg 9 Line 8  Presuming, then, that extended surface water coverage was not an issue at these sites then in terms of pixel removal?*
**RESPONSE:** Sorry but we don't understand this comment.

*Pg 10 Line 9  Perhaps for clarity change Nitrate and ammonium increased... to Nitrate and ammonium concentrations in streamflow increased...  and similarly throughout*
**RESPONSE:** Good point. We have checked the manuscript and changed when needed.

*Pg 10 Line 31  I am wondering back to the initial question on sampling frequency (Pg 5 Line 8) and how the resolution of sampling overlays with this estimate of the fast decay pool (4-20 days). Would more high-frequency sampling during what seems to have been identified as a critical short-term post-fire period yield finer estimates of this critical period length? Further, is it possible that the omission of the first post-fire precipitation event (Pg 5 Line 6) from the sampling design yielded a considerable portion of this fast pool that was unaccounted?*
**RESPONSE:** We agree that there are uncertainties associated with the initial post-fire period, and more sampling points would always be better, but as noted above we did not have the time, access/permits or budget to start sampling sooner, or at higher frequency, with no advance warning of the fire. The fact that some solute peaks occurred after our first sampling visit (in some cases two months later) strongly suggests that we did not miss a major flushing event during the immediate post-fire period. As discussed elsewhere we have added a sensitivity analysis to the results, where we calculate the maximum solute export that could have occured if an earlier peak had occurred (the maximal potential impact on the annual fluvial loss is probably an underestimation of 0.5% for carbon and 1% for nitrogen).

*Pg 11 Lines 2-6  Were these sequences of inequalities statistically assessed? Perhaps including values of each of these peak/baseline ratios here would be informative and a useful normalised metric for other post-fire studies to compare against.*
**RESPONSE:** Good idea and we did consider it ourselves. However, with only 5 catchments a statistical evaluation seems unwarranted.

*Page 11 Line 16  Hanging parentheses*
**RESPONSE:** Thanks for noticing.

*Page 12 Line 5-6  This may be a stretch to generalise from two studies, if no other annual-basis studies of NEE are available.*
**RESPONSE:** Has been reformulated to These values are strikingly similar to our two sites (155 to 165 g C m 2 yr 1 over two years), but further research is needed to establish if such values are typical for boreal uplands post-fire.

*Page 13 Line 2  Is this meant to read first year singular?*
**RESPONSE:** Good catch. Should be first year.

*Page 14 Line 14  an interesting observation on similar impacts from such different types of disturbance what mechanisms would be responsible for these similarities?*
**RESPONSE:** The main reason would be less plant uptake and sometimes in combination of increased mineralisation. This has been added to the discussion.

*Figure 2  perhaps the fire could be delineated as a horizontal line on the figure as similar to Figure 3?*
**RESPONSE:** Added.

*Figure 6  this inclusion of methods/assumptions (text on right of figure) is an excellent contribution to laying out the fluxes in an integrated way such as this.*
**RESPONSE:** Thank you. Even if it has limitation, we think our box diagram helps our understanding of the main post-fire nutrient and carbon flow paths.

**Reviewer 2**

*The impact of wildfire on biogeochemical fluxes and water quality on boreal catchments(Granath et al., 2020, Biogeosciences Discussions) This study reports on the impacts on wildfire on C dynamics and water quality from a boreal forest catchment in Southern Sweden, using paired (before-after) measurements on fire areas. To be honest, I was hoping quite a lot from the paper, as the topic seems really interesting and promising(as both pre fire situation and post-fire conditions were supposed to be included. It would be quite unique possibility to describe quite exactly the C dynamics related withfires (pre fire conditions, combustion, and post-fire conditions), and all this in relatively large scale. Unfortunately, at this stage the paper misses many explanations, and actually entire research is missing some of the needed measurements. Thus, at this stage the authors were not able to convince me that some of their statements are actually valid. At this stage I have the impression, that by leaving the pyrogenic material measurements (charred material, charcoal, ash) out from the research, the authors are overestimating the C losses through combustion. Also, as the water measurements started weeks after fire (and one week after first rain), the authors are underestimating the fluvial C movements. As the authors have not been explaining how they have been using eddy data (they are presenting net ecosystem exchange (NEE) results, that also includes the photosynthesis (carbon uptake), but they havent been explaining the proportions of the photosynthesis and respiration, the authors have not been convincing me that their numbers behind different C fluxes are correct. The authors are also completely ignoring the fact (would expect it at least in discussion) that (at least some of) the areas were logged after fires.*

**RESPONSE:** Thank you for taking the time and performing a very detailed review. We will here respond to the four main listed concerns. These points are partly repeated in the detailed comments but we will sometimes refer to the response here.

Charcoal and other carbon pools: Also reviewer 1 asked about the impact but were less worried that it compromised our conclusions. We do take this issue seriously and have conducted sensitivity analyses (added to the Result section) to better evaluate the effect of our approach. Fortunately,

we have access to data carbon content of the soil charcoal layer. Using data from a recently published study from the same burn (Pérez-Izquierdo et al. 2020 J of Ecology, some of the authors were involved in that study as well) we can conclude that the carbon content is roughly 20-25% in this layer. This is lower than the non-burned organic soil, and the bulk density of this charcoal layer suggests that this layer is less compact as well. Consequently, using non-burned organic soil values for carbon bulk density we likely underestimate carbon loss rather than overestimated as we previously stated in the manuscript. If we assume a ash layer of 1 cm (reported in older pine forest for studied burn [Pérez-Izquierdo et al. 2020 J of Ecology] but for the whole area the thickness is smaller), this underestimation is roughly between 2-45 g C per m2 (or about 0.01-1% of the average calculated loss).

We did not include losses from downed wood as this is a small component in a managed landscape like the one that studied here. The burnt area has around four m3 per hectare of downed wood (Jonsson et al 2016). With a stem density of 412 kg m-3 for Scots pine (Repola 2006), and 50% carbon content, the maximum loss from downed wood is about 80 g carbon per m2 (or circa 1.5% of total C loss). However, this maximum value is very probably unrealistic as downed wood rarely is completely consumed by a fire.

Losses from standing trees were not estimated. It is very hard to make reliable quantifications of such losses (amount of fine branches and needles consumed) and they contribute little to the overall losses in the studied area. We have added a rough estimate of average C storage in branches and needles in the forested area. About 0.5 kg m-2 C is stored in living branches and needles, and 0.15 kg m-2 only in needles. After the fire (charred) needles were still present in the burned crowns and only 21% of the area experienced 100% crown damage. This suggests that losses from living trees likely amounts up to a few percentages of the total C loss in forested areas. We have included this information, and relevant references, in the Methods.

Taken together, in a revised version we provide more information on the uncertainties and provide potential losses associated with the above carbon pools. If anything, our carbon loss estimates are conservative and not an overestimation.

Early hydrological losses: We agree that there are uncertainties associated

with the initial post-fire period, and more sampling points would always be better, but we did not have the time, access/permits or budget to start sampling sooner, or at higher frequency, with no advance warning of the fire. The fact that some solute peaks occurred after our first sampling visit (in some cases two months later) strongly suggests that we did not miss a major flushing event during the immediate post-fire period. We have undertaken a sensitivity analysis of the maximum solute export that could have occured if an earlier peak had occurred (added to the Results section). We describe a sensitivity analysis for the Gärsjöbäcken catchment where we assume that the carbon and nutrient concentration one week after the fire were double the values measured at the first time point (about 3 weeks after). Then the impact on our estimate of the annual fluvial loss is an underestimation of 0.5% for carbon and 1% for nitrogen. This should be viewed as an extreme scenario in our opinion but gives an idea of how small the impact is.

NEE: We focus on NEE as this is the carbon balance and the response of main interest here. We write C emission because NEE showed a net C release. It is clearly important to note that the ecosystem was losing carbon overall *despite* vegetation regrowth during the first three years post-fire; this suggests a large and sustained loss of carbon from soils and dead organic matter. Either the reviewer has misunderstood how NEE data are interpreted here, or we have misunderstood their point. We have edited the text a bit to improve clarity.

Logged areas: We added a new paragraph in the discussion where we discuss potential impacts of salvage logging and what has been reported in the literature. We didn't see a clear effect of salvage logging, which can be expected given the lack of extreme topography. Note that our two focus catchments were not salvage logged (only a few percentages).

*Below are my[reviewer 2] detailed comments:*
*P2 L11: What about Scandinavia? Emissions are bigger or smaller compared to North America, as the fires are completely different in these two regions.*
**RESPONSE:** We are not aware of any data of carbon loss from Scandinavia. Our study is possibly the first one, but we are happy to be corrected.

*P2 L12-13: Compared to what areas? North American areas?Upland soils vs. peatlands?*
**RESPONSE:** Compared to boreal upland soils. This has been clarified.

*P2 L23-24: New study by Rodríguez-Cardona et al 2020(Scientific Reports volume 10, Article number: 8722) shows clear post-fire decrease(although they are using longer chronosequences there).*

**RESPONSE:** Thank you for pointing us to this article. This study has a much longer time-scale (decades), but we see merit in referring to it in the manuscript.

*P2 L24: What is POC export?*

**RESPONSE:** POC=particulate organic carbon. We missed writing out the abbreviation.

*P4 L1: It should be stated somewhere here that (at least some) the areas were logged after fire!*

**RESPONSE:** We are not sure that comparing logged and unlogged areas can be thoroughly tested in our study, and it was never our intention to do so. That is why we did not add it as a separate question/aim. However, we have added a new paragraph in the discussion where this is discussed.

*P4 L1: In intro there is a lot of talk about drained peatlands and/or peatlands.Is this the case also here, are the areas mainly forests on drained peatlands? I think some kind of description of the area would be good to include here.*

**RESPONSE:** We presented data on the percentage of open and forested peatlands in Table 1; total peat cover was around 15-30% and yes, most forested peatlands were drained. The catchment attributes were discussed at the start of the methods, but we notice that this part is lacking a reference to Table 1, and we have added that together with more background information about the area (eg altitude, land-use, forest structure).

*P4 L9: Any expectations/hypothesis?*

**RESPONSE:** We have added text regarding the background and rationale (with a reference).

*P4 L16-17: Would expect more of the area description. How old was the forest? Was it similar through the area or there was many different stands with different age and tree species?*

**RESPONSE:** Study area description was a bit too brief. We have added

more information about the area (eg altitude, land-use, forest structure). In short, forest consisted mostly of even-aged pine dominated stands, varying from clear-cuts up to ¿100 years forest stands.

*P5 L5-8: Can it be that due to late start you have been actually missing some of the C movement (it is washed through different soil horizons with days after rain)?*
**RESPONSE:** See main response at the beginning.

*P5 L28: Is this the same as the "ash layer" mentioned earlier?*
**RESPONSE:** We realise that we used the terms incorrectly. We actually mean the ash layer - sensu Bodi et al., 2014 that uses a broad definition. We have included a definition of ash and changed to ash layer throughout the manuscript.

*P5L30: It can be also up to 60% or even higher (Wiechmann et al 2015. PloS one, 10 (8),e0135014-e0135014). P5 L 30-31: how were the charred logs/snags/stumps treated?If you havent been measuring the pyrogenic carbon (charcoal, ash) separately, you are probably overestimating a lot.*
**RESPONSE:** Thank you for pointing us to Wiechmann et al 2015. However, they report % C of the charcoal particles and not of the charred/ash layer of the organic soil. As we have written in the above response, we now have data on the characteristics of the ash layer, which show a carbon content around 20-25%. Downed wood was not included in our estimates as this component is rather small in these managed forests (see response above regarding downed wood).

*P6 L9-10: Based on Figure1, these transects and sample plot locations are not similar to the burned area. Please specify how these reference transects were located (how far from each other, etc.).*
**RESPONSE:** The transects were chosen to reflect the variation within the burnt area and we believe we succeeded rather well in placing these transects to achieve this goal. Using forest composition and wetness/topography maps we selected similar combinations of forest types, wetness and topography as found across the burnt area. Originally we wanted to model organic soil depth across the landscape but because the mean organic soil layer varied so little between sampling plots and did not correlate with predictors like soil moisture, we decided to use the mean value.

*P6 L14: ...three to five soil cores.... Per transect? Per plot?*
**RESPONSE:** That should be per plot. Thanks for noticing this.

*P6 L19-21: If stated like this, then my question is what about Europe and Scandinavia?*
**RESPONSE:** We are not aware of any studies estimating organic soil loss during fire in northern Europe (but we might have missed studies of course). The method should not be continent-specific and we have reworded this to describe the method in general, rather than pinpoint it to a specific geographical location.

*P7 L4: With eddy, I assume you are measuring net ecosystem exchange (NEE) (including C uptake by photosynthesis and release by respiration). If we assume that everything was killed during fire (but you were saying that at the beginning the fire was not stand replacing) then you would measure the respiration (decomposition, etc.), but the vegetation comes back quite quickly after fire,so I would still say that you are measuring NEE. How you are able to talk about the C emissions? As you are not explaining how you were separate the respiration (C emissions) from the photosynthesis. How big and to what direction is the footprint area of the eddy systems. I would assume that the winds from the west are dominant in these areas, but this way the southern eddy is not measuring fire area (at least most of the time)? Also, the eddys are placed so that you are not able to combine the Closs measurements and eddy data (as they are most probably not overlapping). Any specific reason why the eddys were placed as they were?*
**RESPONSE:** First, we write (P4, L18) that the fire WAS a stand-replacing fire - i.e. everything died more or less. Second, we do indeed focus on NEE as this is the carbon balance and the response of main interest here. We say C emission because NEE showed a net C release. It is important to note that the ecosystem was losing carbon overall *despite* vegetation regrowth during the first three years post-fire; this suggests a large and sustained loss of carbon from soils and dead organic matter.
Third, towers were 2.5 m high. The southern tower is located about five hundred meters from the unburned forest and is indeed measuring only over the burnt area. Fourth, the location of the towers were chosen based on proximity to roads (but still in a closed off area to avoid theft of equipment), representativeness of the area, tree height (tall trees near the tower can fall

over and damage the equipment). We did not know the exact delineation of the catchments when the towers were set up. The fact that the tower happened to be placed just outside the catchment where we did the carbon loss measurements does not invalidate its use for comparing data; we combined the flux tower measurements, the multiple catchment measurements and the distributed soil measurements to seek to understand whole-ecosystem responses to the fire at the landscape scale. To our knowledge, few if any studies have previously obtained such comprehensive data.

*P7 L18-20: This is really big assumption! Taking also into account that you actually havent been taking the formed pyrogenic carbon (charcoal, charred material, etc.) into account (not analyzed it separately), your C loss calculations might be overestimated.*
**RESPONSE:** Regarding pyrogenic carbon, see response above. The assumption seems supported in our view. Erosion is negligible in this system and downwards transportation of carbon particles is likely tiny compared to the amount lost in the fire. Other data on soil carbon that we have collected at some selected sites in the same burnt area did not indicate an increase of carbon in the mineral soil, further strengthening our assumption that changes in carbon stock can be ascribed to gaseous emissions (Pérez-Izquierdo et al. 2020 J of Ecology).

*P9 L13-15: So you are saying that 95% of the C emitted during the fire was coming from O-horizon? You had high severity, stand replacing fires on areas (high intensity), all the trees killed, vegetation removed, and then more than 95% comes from O-horizon? On table 2 there is only one value for emissions during the fire, and no separation by vegetation and/or soil.*
**RESPONSE:** In boreal forests the organic soil is a large carbon pool, and most of it was combusted during the fire. As mentioned above, we did not include downed wood in our estimates, but this pool is very small in managed forests. We also did not estimate loss from trees (needles, fine branches). The 95% statement was comparing belowground and the forest floor. This will be corrected and we have included the potential contribution from downed wood and trees in the manuscript (see earlier response). Table 2 only gives the sum as we wanted to focus on the overall picture. However as the review has queried this interpretation we can provide disaggregated estimates of carbon loss in the text.

*P9 L23-25: I still think that you were actually missing the biggest fluvial losses (the pyrogenic material that is washed away with first rain event).*
**RESPONSE:** See our response earlier.

*P10 L1-2: Base on the figures you have been measuring NEE with eddy. Unfortunately, there is no data available about vegetation recovery (biomass, coverage), but I have the impression that 3 years after fire, there is already some new vegetation also in areas with high severity. So one cant talk anymore about C loss when interpreting the NEE values.*
**RESPONSE:** Yes, for sure there was vegetation recovery in the flux tower footprints, and throughout the study area. The flux towers measured this as part of the NEE, i.e. the measurements represent the balance of vegetation carbon gain and ongoing soil and biomass carbon loss. These measurements clearly show sustained positive NEE over the 3 years post-fire, i.e. the ecosystem was a net source of CO2 to the atmosphere despite vegetation regrowth. Either the reviewer has misunderstood how NEE data are interpreted here, or we have misunderstood their point. We have done a few edits that hopefully have added clarity on this.

*P10 L4-5: Now the talk is about C uptake (my previous comment). But the vegetation regrowth data is not presented, and it is still not explained how you separated the respiration and uptake data from each other.*
**RESPONSE:** Vegetation growth is presented as leaf area increase. We are not sure we follow the reviewers comment here as we do say net carbon uptake, i.e. the balance of vegetation growth and soil/dead biomass loss as noted above. We did not intend to separate uptake and respiration in our presentation of the data.

*P11 L11-12: Sorry, but based on your results and talk, Im not convinced! By not taking into account (analyzing separately) the amount of charred material and charcoal, you are overestimating the direct emissions from the fires.*
**RESPONSE:** See earlier response.

*P11 L16-18: Sorry, but you missed the first rain event (if the first samples were taken week after first post-fire rain), and with that probably also DOC that was washed away (or washed to deeper soil horizons) from the areas. So, I assume you are underestimating the fluvial C loss.*

**RESPONSE:** See earlier response on the early post-fire period.

*P14 L23: Discussion is completely missing the fact that the areas(at least some of them) were logged after fire. How the mixing of soil (pyrogenic material and soil) by machinery would affect the emissions and water quality? How the logging (removing the material that would start to decompose on areas) could affect the C fluxes?*

**RESPONSE:** It is correct that a large portion of two catchments were logged, and one other catchment experienced minor logging. Only older stands were salvage logged. Note that the two focus catchments were not logged (except a tiny part on one edge of the catchment and along some roads). Logging was mainly done between 4-12 months after the fire (in the manuscript it said that logging was done within 3-6 months after the fire, but this is not completely accurate). Interestingly, when examining the water chemistry the logged catchments do not stand out from the other catchments. There is likely not an impact due to the absence of extreme topography. Removal of singed older trees probably had little impact on carbon emission at the site over the first years. This is also what has been reported in other studies. While the trees were killed, most of the stemwood remained intact after the fire (and most trees that were rooted into mineral soil remained standing). This woody material is slow to decompose (particularly when singed), and (in areas that were not salvage logged) it was still present by the end of our study period. The gradual decay of this material and charred needles will have contributed to measured NEE in these areas, which our results suggest made a relatively modest contribution to overall carbon loss during the three years of measurements. Clearly, decomposition of dead biomass will continue to contribute to CO2 loss for many years to come, so the proportional contribution of this C pool to total losses might be expected to gradually increase over time; we have noted this in the revised manuscript. Where salvage logging occured it is clearly a more rapid and substantial pathway for C loss as the wood is removed. But again, our two focus catchments did not experience logging. A paragraph discussing the potential impact (or lack of) has been added in the Discussion section.

---

## Author Response (AR2)

**Reviewer 2**

*Although, the authors have been stating that they have included the text regarding the background and rationale, I would still like to see some more expectations/hypothesis at the end of introduction (as based on this one can also see, why the study was done at all).*
RESPONSE: This was also pointed out by the editor. We dont think adding hypotheses retroactively is the best option here. Instead we re-wrote the "aims" paragraph and stated clear study objectives, which should help the reader to understand why we did the study. We believe this is a improvement compared to the previous detailed list of measurements and approaches, and it fits well with the current structure of the manuscript.

*In discussion (new text dealing with logged vs. non-logged areas), while stating "…, but previous studies have not found clear evidence of increased soil C losses compared to unlogged areas", I would not use reference from Mediterranean areas, as there are similar comparisons done also in Northern-Europe (for example: Parro et al., 2019, Journal of Environmental Management, 233, 371-377).*
RESPONSE: Good point. We removed the reference and instead added Parro et al 2019.

---

## Author Response (AR3)

*I noticed that you switch between "burned" and "burnt" repeatedly.*
RESPONSE: We have changed to burnt to burned throughout the manuscript.